# Imagine While Reasoning in Space: Multimodal Visualization-of-Thought

**Chengzu Li** [* 1 2 †]  **Wenshan Wu** [* 2]  **Huanyu Zhang** [2 3 †]  **Yan Xia** [2]
**Shaoguang Mao** [2]  **Li Dong** [2]  **Ivan Vulić** [1]  **Furu Wei** [2]

## Abstract

Chain-of-Thought (CoT) prompting has proven highly effective for enhancing complex reasoning in Large Language Models (LLMs) and Multimodal Large Language Models (MLLMs). Yet, it struggles in complex spatial reasoning tasks. Nonetheless, human cognition extends beyond language alone, enabling the remarkable capability to think in both words and images. Inspired by this mechanism, we propose a new reasoning paradigm, **Multimodal Visualization-of-Thought (MVoT)**. It enables visual thinking in MLLMs by generating image visualizations of their reasoning traces. To ensure high-quality visualization, we introduce *token discrepancy loss* into autoregressive MLLMs. This innovation significantly improves both visual coherence and fidelity. We validate this approach through several dynamic spatial reasoning tasks. Experimental results reveal that MVoT demonstrates competitive performance across tasks. Moreover, it exhibits robust and reliable improvements in the most challenging scenarios where CoT fails. Ultimately, MVoT establishes new possibilities for complex reasoning tasks where visual thinking can effectively complement verbal reasoning.

## 1. Introduction

Chain-of-Thought (CoT) prompting has substantially enhanced the reasoning capacity of Large Language Models (LLMs) (Jiang et al., 2023; OpenAI, 2023; Dubey et al., 2024). By generating explicit reasoning traces, CoT enables the models to articulate their thought processes step-by-step. This advancement has enabled step-by-step mathematical

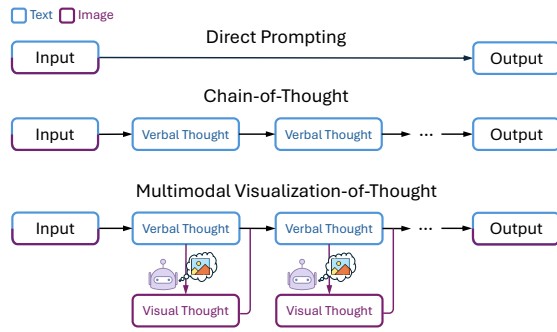

*Figure 1.* Multimodal Visualization-of-Thought (MVoT) enables Multimodal Large Language Models to generate interleaved reasoning traces across different modalities. While traditional CoT relies solely on verbal thought, MVoT facilitates visual thought to visualize the reasoning traces. This reasoning paradigm resembles human cognition to think in words and images seamlessly.

reasoning, logical deduction (Lam et al., 2024), and advanced problem-solving capabilities (Zhang et al., 2024a). However, its performance deteriorates significantly when confronted with complex spatial reasoning tasks (Wang et al., 2024a; Li et al., 2024b; Ray et al., 2024).

Recent research efforts have extended CoT to *multimodal* models through two primary approaches. The first approach utilizes two-stage strategies that initially extract image information through methods such as captioning (Zhang et al., 2024b), scene-graph generation (Mitra et al., 2024), or bounding-box detection (Lei et al., 2024) before conducting reasoning. The second approach implements ReAct-style pipelines (Yao et al., 2023) that leverage external tools such as code interpreters or specialized vision models (Yang et al., 2023b; Hu et al., 2024; Zhou et al., 2024b), to obtain image observations from the environment. While these pipelines successfully handle both text and image input, they remain heavily dependent on separate visual modules or external toolsets. This dependency complicates their adaptation to advanced and more complex spatial reasoning tasks.

Human cognition, however, transcends language (and text) alone, demonstrating the capacity to think in both words and images seamlessly. The dual-coding theory (Paivio,

*Equal contribution [1]Language Technology Lab, University of Cambridge [2]Microsoft Research [3]Institute of Automation, Chinese Academy of Sciences. Correspondence to: Chengzu Li <cl917@cam.ac.uk>.

*Proceedings of the 42nd International Conference on Machine Learning*, Vancouver, Canada. PMLR 267, 2025. Copyright 2025 by the author(s).

1991) and working memory model (Baddeley, 1992) explain this phenomenon, positing that humans process information through both verbal and non-verbal channels. This dual processing capability is crucial for reasoning and guides decision-making. For instance, humans naturally manipulate mental images to understand the physical world and conceptualize visual outcomes (Moulton & Kosslyn, 2009).

In the context of spatial reasoning, similar deficiency of LLMs and MLLMs, thinking in a single text modality (termed *verbal* henceforth) has been observed. Based on verbal CoT, the Visualization-of-Thought (VoT) (Wu et al., 2024b) elicits the reasoning process by introducing *textual 'visualizations'* as mental images in spatial reasoning tasks. Put simply, VoT builds visualizations via simplified, text proxies. Besides this too simplistic representation of visualization, another significant limitation, which holds both for CoT as well as for VoT, is the dependence on purely textual representation of the reasoning paths. This reliance becomes problematic in complex multimodal tasks, where textual representations often fail to capture the intricate visual patterns and spatial layouts of images (Hu et al., 2024). Users frequently find it challenging to interpret the reasoning process without clear and intuitive visual illustrations that complement textual representation. Furthermore, it remains unclear whether MLLMs can engage in genuine reasoning within a visual space while thinking with textual utterances. Given the reasons above, we pose a key question: *can MLLMs imagine in visual modality while reasoning?*

In parallel, recent research has expanded beyond multimodal understanding of visual inputs to include multimodal generation, where foundation models can also produce outputs in the visual modality. This advancement has led to the development of sophisticated systems such as Chameleon (Chameleon Team, 2024), Transfusion (Zhou et al., 2024a), LatentLM (Sun et al., 2024b) and Janus-Pro (Chen et al., 2025). These *multimodal-native* models demonstrate proficiency in both interpreting and producing high-quality outputs across textual and visual domains. The emerging capability of multimodal generation opens new possibilities for extending verbal reasoning to native visual thinking, enabling the visualization of reasoning traces through images.

Building upon these advancement, we propose **Multimodal Visualization-of-Thought (MVoT)**. It leverages multimodal-native architectures to transcend the text-form thinking into multimodal native reasoning through generating image visualizations of their reasoning traces. This reasoning paradigm enables the model to 'think' in words and images in combination seamlessly, while avoiding the potential errors being introduced in captioning the images. By incorporating native visual thought during reasoning process, MVoT offers more straightforward illustrations of the reasoning process and simultaneously enhances both reasoning quality and model interpretability.

Specifically, in this work, we implement MVoT through fine-tuning an established autoregressive MLLM: Chameleon-7B (Chameleon Team, 2024). To enhance visualization quality during reasoning, we introduce **token discrepancy loss** that bridges the gap between separately trained tokenizers. We validate MVoT's effectiveness through controlled experiments across three dynamic spatial reasoning tasks. MAZE (Ivanitskiy et al., 2023) and MINIBEHAVIOR (Jin et al., 2023) focus on interactions with spatial layouts. FROZENLAKE (Brockman, 2016) emphasizes fine-grained pattern representations in dynamic environments. Experimental results demonstrate that MVoT achieves competitive performance across tasks, outperforming traditional CoT by over 20% in challenging scenarios.

The main contributions of this paper include:

- We propose **Multimodal Visualization-of-Thought (MVoT)**, a multimodal native reasoning paradigm that unifies text and vision within the reasoning traces. To our knowledge, it's the first to naturally generate visual thought as part of the reasoning process. It establishes new possibilities for complex tasks where visual thinking effectively complements verbal reasoning.
- We implement MVoT in Chameleon-7B and introduce **token discrepancy loss** in auto-regressive MLLM to bridge the gap between separately trained tokenizer.
- We conduct comprehensive experiments and ablation studies across three spatial reasoning tasks with newly collected datasets, demonstrating that MVoT exhibits superior adaptability and robustness compared to CoT in complex scenarios.

## 2. Related Work

**Multimodal Chain-of-Thought Reasoning** Chain-of-Thought (CoT) (Wei et al., 2022) prompting has considerably enhanced the reasoning capacities of LLMs. To adapt CoT for multimodal models, recent research has explored various methodologies. Some investigations adopt a two-stage approach, where image information is initially transformed and grounded into text (Zhang et al., 2024b), graph structure (e.g., scene graphs (Mitra et al., 2024) or knowledge graphs (Mondal et al., 2024)), or bounding boxes (Lei et al., 2024) before reasoning. Other studies use ReAct-style pipelines (Yao et al., 2023) that integrate external tools to process and reason with image observations. These tools include code interpreters and specialized vision models (Yang et al., 2023b; Hu et al., 2024; Zhou et al., 2024b; Gao et al., 2024). Although these approaches effectively manage both textual and visual inputs, they rely heavily on separate visual modules or toolsets which limits the expressiveness of the methods. To address these limitations, we propose MVoT, a novel reasoning method designed to leverage multimodal-

native understanding and generative capabilities. MVoT enables the generation of interleaved reasoning traces across multiple modalities, providing an integrated and flexible approach to multimodal reasoning with better interpretability and more robust reasoning quality.

**Multimodal Spatial Reasoning** Multimodal spatial reasoning involves understanding and reasoning about the spatial relationships among objects, their movements, and interactions with the environment (Chabris et al., 2006; Li et al., 2024a; Newcombe, 2024; Zhang et al., 2025). Despite advancements achieved with LLMs and MLLMs, this remains a significant challenge and attracts growing research interest (Achiam et al., 2023; Yang et al., 2024; Ramakrishnan et al., 2025). To systematically evaluate multimodal spatial reasoning capabilities, several benchmarks have recently been introduced, covering diverse perspectives and tasks (Liu et al., 2023a; Wang et al., 2024a; Li et al., 2024b; Ray et al., 2024). However, few research has touched base on the interplay between actions and changes in spatial layout (Li et al., 2024b; Wu et al., 2024a), which requires dynamic imagination and tracking the states as actions alter the environment. Various approaches have been proposed to tackle the challenges associated with spatial reasoning for MLLMs and LLMs: SpatialVLM (Chen et al., 2024) and SpatialRGPT (Cheng et al., 2024) improve multimodal spatial reasoning by leveraging 3D VQA or scene graph data for supervised finetuning. VoT (Wu et al., 2024b) proposes a novel prompting approach by introducing textual imagery representation to facilitate dynamic reasoning for LLMs. Despite these efforts, existing methods fail to unlock the inherent reasoning capabilities within the multimodal-native models to imagine the spatial dynamics. Our approach unifies text and vision within the reasoning traces and improve the interpretability and robustness by generating mental images aligned with spatial reasoning.

# 3. Multimodal Visualization-of-Thought

Humans often create mental images to inform decision-making. Rather than relying on verbal thoughts as text proxies to mimic these mental images, MVoT enables models to reason in multimodality by generating image visualizations as their visual thoughts. By combining thoughts in both modalities, this novel reasoning paradigm offers a more intuitive and effective way to elicit the multimodal reasoning process with enhanced expressiveness.

## 3.1. Formulation

We formulate the process of MVoT as follows. Given a multimodal input sequence, the model is expected to generate interleaved multimodal thoughts as part of the reasoning process and ultimately produce a final answer. Let $\mathcal{P}_\theta$ rep-

resent a pre-trained MLLM with parameters $\theta$, $x$ denote a multimodal input sequence, $z$ and $v$ a language sequence of verbal thoughts and an image sequence of visual thoughts.

In multi-hop spatial reasoning tasks with input $x$, CoT prompting generates intermediate steps $\hat{z}_1, \cdots, \hat{z}_m$, where each $\hat{z}_i$ is sampled sequentially based on the inputs and previous generated steps. The final output is concluded based on all prior steps. **MVoT** enhances this process by adding a image visualization $v_i$ to each intermediate step $z_i$, then the subsequent step $z_{i+1}$ is sampled conditioned on prior steps $\hat{z}_1 \cdots \hat{z}_i$ and visualizations $\hat{v}_1 \cdots \hat{v}_i$, as shown in Figure 1.

As defined in the Equation 1 and 2, it forms interleaved reasoning traces and image visualizations.

$$\hat{v}_i \sim \mathcal{P}_\theta(v_i \mid \hat{z}_1, \hat{v}_1, \cdots, \hat{v}_{i-1}, \hat{z}_i) \tag{1}$$

$$\hat{z}_{i+1} \sim \mathcal{P}_\theta(z_{i+1} \mid x, \hat{z}_1, \hat{v}_1, \cdots, \hat{z}_i, \hat{v}_i) \tag{2}$$

To empower MLLMs with MVoT capabilities, we train the model on multimodal inputs $x$ and their corresponding output labels, which include multimodal rationales $z_1, v_1 \cdots z_n, v_n$ and the final answer. This training strategy enables the model to learn interleave verbal reasoning steps and corresponding visual thoughts, enhancing its ability to handle complex reasoning tasks that require thinking in multimodality.

## 3.2. Training with Autoregressive MLLMs

In this section, we focus on autoregressive MLLMs with discrete image tokens for both training and inference. However, as a reasoning paradigm, MVoT can be extended to other model architectures and modalities, provided they meet the requirements for interleaved multimodal generation.

**Multimodal Sequence Modeling** As shown in 3, we follow the architecture of Chameleon (Chameleon Team, 2024), which leverages a unified Transformer to process both image and text tokens. The architecture integrates two tokenizers: an image tokenizer based on Esser et al. (2021) and a text tokenizer, which convert images and text into discrete token sequences, respectively. The image tokenizer uses a discrete codebook to encode input images into a sequence of image tokens, while the text tokenizer maps textual data into corresponding token sequences. These token sequences are concatenated and processed by a causal transformer model.

**Notation** We denote MLLM's codebook as $\mathcal{C} \in \mathbb{R}^{N \times D}$, where $N$ is the number of codebook entries, and $D$ is the dimensionality of codebook entries. Let $t_{\text{vis}}$ and $e_{\text{vis}}$ denote the visual codebook indices and embeddings. The predicted values are indicated with a hat symbol.

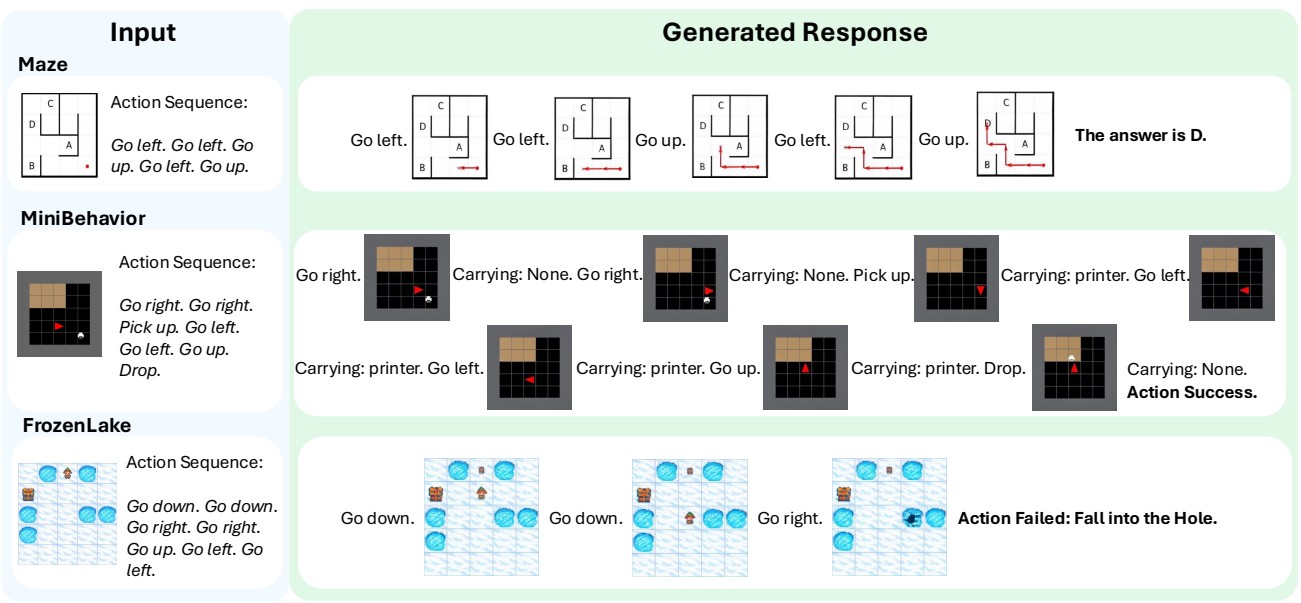

*Figure 2.* Illustrations of MVoT reasoning process. Interleaved verbal thoughts and visual thoughts are generated by MLLM seamlessly.

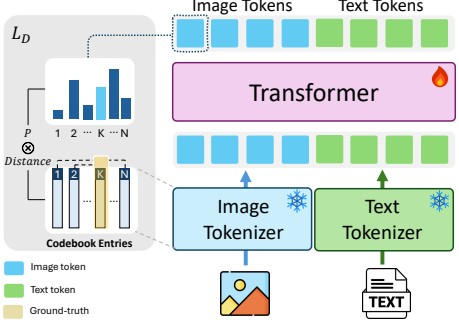

*Figure 3.* MVoT training with token discrepancy loss $\mathcal{L}_D$.

**Token Discrepancy Loss** While language modeling unifies both text tokens and image tokens within a single autoregressive model, the discrepancy between these separately trained tokenizers can degrade the quality of generated images. To mitigate this issue, we introduce token discrepancy loss into the autoregressive MLLM architecture, as shown in Figure 3. This loss design bridges the gap between language modeling and visual embedding space while ensuring that gradients remain intact.

Token discrepancy loss $\mathcal{L}_D$ minimizes the discrepancy between the predictions and labels in visual embedding space. Equation 3 and 4 illustrate the token discrepancy loss $\mathcal{L}_D$. To capture relationship among image tokens, we first compute the similarity matrix $\mathcal{S} \in \mathbb{R}^{N \times N}$. The similarity between $t_{\text{vis}_i}$ and other image tokens is measured using their pairwise distances in the visual embedding space. Specifically, the similarity is determined by the mean squared error

(MSE), as described in Equation 3, where larger distances indicate lower similarity.

$$\mathcal{S}_{t_{\text{vis}_i}} = [\text{MSE}(e_{\text{vis}_i}, e_{\text{vis}_1}), \cdots, \text{MSE}(e_{\text{vis}_i}, e_{\text{vis}_N})] \quad (3)$$

The model predicts the probability distribution $\text{P}(t_i) \in \mathbb{R}^{1 \times N}$ for the $i$-th image token over the image token vocabulary. $\mathcal{L}_D$ penalizes probabilities assigned to tokens that deviate significantly from their corresponding label $t_{\text{vis}_i}$ in the visual embedding space. By aligning the visual embeddings of predictions with those of the ground-truth tokens, $\mathcal{L}_D$ aims to enhance the quality of the generated images.

$$\mathcal{L}_D = \sum_{i=1}^{n} \mathcal{S}_{t_{\text{vis}_i}} \cdot \text{P}(t_i) \quad (4)$$

**Training** The causal Transformer model is fine-tuned using the next-token prediction objective, while the image tokenizer and text tokenizer are kept frozen throughout the process. With token discrepancy loss $\mathcal{L}_D$ for image tokens, and the cross-entropy loss $\mathcal{L}_C$ for both text tokens and image tokens, the loss function is described as follows.

$$\mathcal{L} = \mathcal{L}_C + \mathcal{L}_D \quad (5)$$

## 4. Spatial Reasoning Tasks

Motivated by Ray et al. (2024), we select three dynamic reasoning tasks in space that requires the model to dynamically locate objects, understand how the environment evolves, and predict outcomes when actions are ap-

plied. To assess the MLLMs with controllability for analysis, we ground these spatial reasoning tasks within grid-based environments: MAZE navigation (Ivanitskiy et al., 2023), InstallingAPrinter from MINIBEHAVIOR (Jin et al., 2023) and FROZENLAKE simulation (Brockman, 2016; Wu et al., 2024a). MAZE navigates in abstract mazes and MINIBEHAVIOR includes the spatial properties of the objects and interactions with the environmental layouts, while FROZENLAKE contains fine-grained pattern details instead of abstract symbols. The tasks encompass various levels of controlled complexity as the abstraction of the real world.

### 4.1. MAZE

In maze navigation, the model is provided with an initial image describing the maze layout and the starting position of the agent in the maze. The agent should navigate through the maze and reach the destination. In our work, we define the MAZE task as follows. Given an initial maze with a starting point and a sequence of actions, the model is supposed to follow the actions and predict the final destination chosen from the locations labeled A, B, C or D in the maze.

### 4.2. MINIBEHAVIOR

We select InstallingAPrinter from MINIBEHAVIOR (Jin et al., 2023) embodied AI tasks as a representative scenario for our experiments. It requires the agent to first locate the printer on the floor, pick it up and carry it to the table to toggle it on. As shown in Figure 2, the printer is represented by a small printer symbol, the agent is represented by a red triangle and the table is represented by a brown area. Aligned with previous settings, we define the MINIBEHAVIOR task in our work as follows. Given a sequence of actions and an environment layout in image, the model should predict the outcome of conducting the actions. The outcomes include whether the agent successfully executes tasks, such as picking up the printer or placing it on the table, and whether objects are missing from the environment. This task expands the action space of MAZE by introducing the interaction with the environment, while maintains a similar level of perception difficulty with simple symbolic representations.

### 4.3. FROZENLAKE

FROZENLAKE is initially proposed by Wu et al. (2024a) implemented with Gym (Brockman, 2016), which is similar to maze navigation but with more complex patterns and details. As shown in Figure 2, it simulates a grid-based frozen lake. The agent is supposed to start from the designated position and reach the destination safely without falling into the 'holes'. Based on the spatial reasoning task from Wu et al. (2024a), we define the FROZENLAKE task as follows. Given a sequence of actions and the grid-based

frozen lake layout with the start and goal position, the model has to determine the consequence of following given actions. Compared to MAZE and MINIBEHAVIOR, FROZENLAKE contains more diverse image details and its environment is more complex considering the number of key entities such as holes.

## 5. Experiments

### 5.1. Experimental Setups

**Data.** We construct datasets for three spatial reasoning tasks, as described in Section 4, encompassing varying levels of complexity in patterns and action spaces. The dataset statistics are presented in Table 4. Detailed information on data collection is provided in App. B. We structure the data as interleaved text-image pair to train MVoT, as in Section 3.

**Model and Experiments.** We use Anole-7B (Chern et al., 2024) model as the backbone. Anole is tuned on Chameleon (Chameleon Team, 2024) and can generate interleaved text and image, making it well-suited for MVoT. We only tune part of the model's parameters with LoRA (Hu et al., 2021) in an instruction tuning manner (Liu et al., 2023b) on MI300X for 40 epochs, where only the loss from the predictions is optimized. Additionally, we evaluate the performance of GPT-4o (OpenAI, 2024) with zero-shot inference, CoT and MVoT in the ReAct-style pipeline (Yang et al., 2023b). Detailed prompting templates and hyperparameters for each task and system variant are provided in App. C.

We compare the MVoT with the following families of system variants: 1) Direct Prompting (*Direct*): The model directly outputs the choice index without intermediate reasoning. 2) Chain-of-Thought (*CoT*): The model is instruction-tuned to reason step-by-step, incorporating coordinates and environment layout described in text, before concluding with the final answer. 3) Training with Interleaved Text-Image Pairs (*Interleaved*): This method follows the standard training approach for MLLMs, interleaving text and image data. However, *Interleaved* calculates the loss only on text tokens while excluding image tokens. In contrast, MVoT computes the loss across all token types. The comparisons among these system variants are summarized in Table 1.

**Metrics** We extract the predicted answer from model output by pattern matching with 'the answer is'. Accuracy for multiple-choice question answering is calculated by comparing the predicted choice with the ground-truth answer.

### 5.2. Experimental Results

**MVoT outperforms *Direct* and GPT-4o with interpretability.** Experimental results across all three simulation tasks reveal that *Direct* struggles with overfitting to spatial

*Table 1.* Experimental results from different system variants across all tasks. *Coord* denotes whether the system is instructed to use textual coordinates during inference; *Layout* denotes whether the system first caption the environment with text for further inference; *Image* refers to whether the images of intermediate states are visible to the model during training. Best and second best performance are illustrated with corresponding colors. Methods with underlines are fine-tuned on our datasets. ↓ indicates worse performance than *Direct* baseline.

| Model | Method | Training Variants | | | Output | Task | | |
|---|---|---|---|---|---|---|---|---|
| | | *Coord* | *Layout* | *Image* | | MAZE | MINIBEHAVIOR | FROZENLAKE |
| GPT-4o | Zero-Shot Direct | - | - | - | Text | 0.7100 | 0.4576 | 0.4976 |
| | Zero-Shot CoT | - | - | - | Text | 0.7386 | 0.4676 | 0.4664 |
| | With Visual Thought* | - | - | - | Text | 0.8556 | 0.6440 | **0.8021** |
| Anole 7B | Direct | ✗ | ✗ | ✗ | Text | 0.7171 | 0.7250 | 0.7788 |
| | CoT | ✓ | ✓ | ✗ | Text | **0.9792** | **0.9812** | 0.6148 ↓ |
| | - w/o layout | ✓ | ✗ | ✗ | *Text* | *0.7023 ↓* | *0.6000 ↓* | *0.5974 ↓* |
| | Interleaved | ✗ | ✗ | ✓ | Text | 0.8678 | 0.8406 | 0.6460 ↓ |
| | **MVoT** | ✗ | ✗ | ✓ | Text, Image | **0.9295** | **0.9514** | **0.8560** |

\* Visual thought is generated by Anole 7B implemented with MVoT.

reasoning tasks, achieving an accuracy of approximately 70%. GPT-4o performs even worse, both with and without CoT prompting, underscoring the inherent difficulty of these reasoning tasks. In contrast, MVoT demonstrates consistent improvements. It surpasses *Direct* by 7% on FROZEN-LAKE and achieves over 90% accuracy on both MAZE and MINIBEHAVIOR. Beyond its superior performance, MVoT also provides verbal and visual thoughts of intermediate reasoning states. This feature enhances interpretability, offering a clearer and more intuitive understanding of its reasoning process compared to *Direct*.

**MVoT achieves comparable or better performance than CoT with enhanced robustness.** CoT achieves over 95% accuracy on the MAZE and MINIBEHAVIOR by explicitly describing the environment layout and agent location with textual coordinates. However, it performs worse than *Direct* baseline on FROZENLAKE. In contrast, MVoT demonstrates comparable performance on MAZE (92.95%) and MINIBEHAVIOR (95.14%), while achieving a higher accuracy of 85.60% on FROZENLAKE compared to both *Direct* and CoT. This demonstrates better stability and robustness of MVoT than CoT.

Although CoT desmonstrates strong results on MAZE and MINIBEHAVIOR, it shows **vulnerabilities and limitations**:

*(1) CoT is sensitive to environment complexity.* CoT underperforms on FROZENLAKE, where the environment is more complex due to the presence of additional key entities (e.g., holes), compared to the other tasks. Error analysis reveals that 90.80% of CoT's mistakes on FROZENLAKE stem from inaccurate coordinate descriptions of holes in the environment. An illustration of this type of error is presented in Figure 9 in the Appendix. In contrast, no such errors are observed in MAZE and MINIBEHAVIOR. Furthermore, CoT's performance on FROZENLAKE deteriorates as grid size increases, dropping from 0.9401 on a $3 \times 3$ grid to 0.3911 on

a $6 \times 6$ grid, as detailed in Table 15 in the Appendix.

*(2) CoT relies heavily on textual description of the environment.* CoT performs well on MAZE and MINIBEHAVIOR when the environment layout is accurately described through text. Captioning the environment layout simplifies the task into a textual reasoning process, eliminating the need for visual references. However, as shown in Table 1, when reasoning is performed using only the agent's coordinates without explicit textual descriptions of the environment, CoT consistently underperforms the *Direct* baseline. This reliance is particularly evident in FROZENLAKE, where flawed predictions stem from inaccurate environment descriptions. These limitations constrain the generalization and reliability of CoT, particularly in complex environments.

Meanwhile, MVoT doesn't have the ineffectiveness above.

*(1) MVoT is more robust to environment complexity compared to CoT.* MVoT maintains stable performance across varying grid sizes within each task, as shown in Table 15 in the Appendix. In FROZENLAKE, even as the environment becomes more complex with larger grid sizes and increased number of 'holes', as shown in Table 7, MVoT consistently achieves over 83% task performance. In contrast, CoT shows a significant decline in performance as environmental complexity rises, highlighting the robustness of MVoT in handling more challenging scenarios.

*(2) MVoT demonstrates better interpretability than CoT.* Rather than solely relying on textual descriptions, MVoT elicits reasoning process by visualizations, effectively mitigating potential errors introduced by inaccurate text-based captions for complex environments. Moreover, visual thought provides a more direct and interpretable way to track the reasoning process compared to aligning textual coordinates within images. This calls for the need of incorporating multimodal reasoning, leveraging both textual and other modalities, such as vision, rather than reasoning solely

*Table 2.* Performance upper bounds achieved by combining predictions from CoT and MVoT across three tasks.

|  | MAZE | MINIBEHAVIOR | FROZENLAKE |
|---|---|---|---|
| CoT | 0.9792 | 0.9812 | 0.6148 |
| MVoT | 0.9295 | 0.9514 | 0.8560 |
| Upperbound | 0.9984 | 1.0000 | 0.9246 |

in text as the primary modality.

**Learning from interleaved multimodal rationales for better and grounded reasoning.** Data-wise, our findings suggest that incorporating interleaved training data, even without generating visualizations, improves reasoning performance. Method-wise, MVoT achieves higher and more consistent improvements across all tasks. Compared between *Direct* and *Interleaved*, we observe that the inclusion of interleaved visualizations during training, despite not contributing directly to the loss, improves performance on MAZE and MINIBEHAVIOR by over 10%. This empirically indicates that these interleaved images helps the model in leveraging visual cues for reasoning during optimization. However, despite the advantages of the *Interleaved* paradigm, we witness a drop in task performance on FROZENLAKE. We conjecture that this may be attributed to the complexity of the visual cues in FROZENLAKE for *Interleaved*. This highlights the unique challenges of the FROZENLAKE task and illustrates the limitations of the traditional *Interleaved* training paradigm. In contrast, when interleaved multimodal rationales serve as supervision signals for visual tokens as in MVoT, the model explicitly grounds its reasoning by generating visualizations, leading to improved performance across all tasks. These findings provide valuable insights into leveraging interleaved multimodal thoughts to enhance reasoning with MLLMs and emphasize the need for further research.

**Equip Proprietary Models with MVoT** In addition to use a single 7B open-sourced model to generate multimodal thought and conclude the answer, MVoT also provides flexibility of being used as plug-ins for other proprietary models including those accessible via APIs. We provide GPT-4o with the visual thoughts from fine-tuned MVoT model after GPT-4o generates the verbal thought. We witness an improvement in performance by over 15% accuracy across all the tasks as shown in Table 1. We hope this work inspires further exploration and fosters advancements in multimodal agent reasoning with multimodal thoughts.

**MVoT Complements CoT in Reasoning.** To investigate whether MVoT and CoT share similar reasoning capabilities and fail on the same data, we combine their predictions and calculate the upper-bound performance. In this setting, a data point is considered correct if either MVoT or CoT generates the correct prediction. Table 2 shows the upper-

bound performance reaches nearly 100% accuracy on Maze and MINIBEHAVIOR, and 92% accuracy on FROZENLAKE. These findings suggest that MVoT complements CoT by offering an alternative reasoning strategy, enabling the ensemble of approaches to further enhance performance.

In summary, MVoT demonstrates its effectiveness in performance, better generalization than CoT in eliciting the reasoning state during spatial reasoning.

## 6. Discussions and Ablations on Visualization

During training, the model generates the next visual thought based on the previous golden image. On the other hand, MVoT recursively generate multimodal thoughts (texts and image visualizations) based on the previously generated thoughts, illustrating the difference between paradigms. We refer to these two approaches as 'image editing' and 'MVoT' in the following discussion. Given that MVoT operates through recursive generation, our focus in this section is primarily on discussing the visualization quality of MVoT.

**Qualitative Analysis** Figure 4 illustrate examples of correct and incorrect generated images for FROZENLAKE. More generated visualization are shown in Figure 6 and 7 in Appendix D.2. We classify errors of visualization generation into the following categories: (1) Wrong Visualization: The generated visualization is inaccurate. (2) Redundant Patterns: Unnecessary or irrelevant patterns are visualized outside the intended area for modification. Furthermore, in the FROZENLAKE task, we observe that generated image details often blur as pattern complexity increases compared to MAZE and MINIBEHAVIOR. Patterns, such as background details in FROZENLAKE MVoT visualizations, often show minor inconsistencies between generated visual thoughts. Similar differences are also noted between original images and reconstructed images through tokenization and detokenization. The variability frequently results in a loss or perturbation of fine-grained details and highlights limitations in the expressiveness of MLLMs. These findings underscore the need for further research to improve the fidelity of image tokenization and generation in autoregressive MLLMs.

**Quantitative Metrics** To evaluate the quality of generated visual rationales, we define automatic evaluation metrics based on the identified types of errors:

- Visualization Accuracy (V-Acc.): Measures the accuracy of visualizing the intended modifications in the grid corresponding to the next action.
- Visualization Pattern Redundancy (V-Red.): Assesses the presence of unintended visual patterns in regions outside the targeted area of modification.
- Visualization Correctness Step (V-Steps): the average length of first k consecutive correct visualizations within an action sequence.

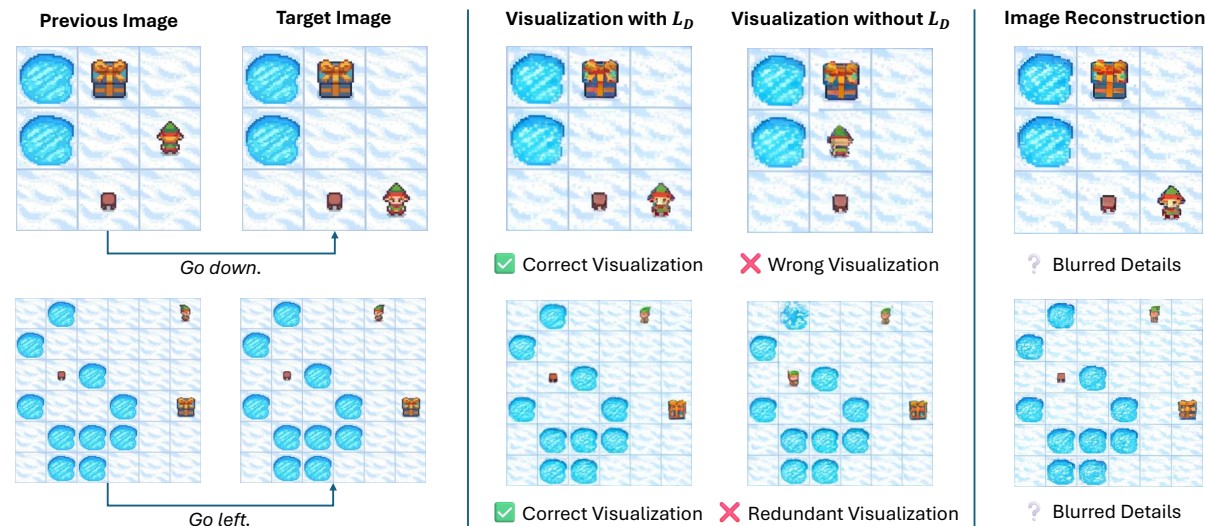

*Figure 4.* Qualitative analysis illustrating: 1) FROZENLAKE visualization quality by comparing systems trained with and without token discrepancy loss ($\mathcal{L}_D$); and 2) differences in reconstructed images using the image detokenizer on tokenized inputs.

*Table 3.* Quantitative metrics for MVoT visual thoughts with or without token discrepancy loss $\mathcal{L}_D$. Best results are highlighted in **bold**. Metrics with ↑ indicate that higher values mean better performance and vice versa.

| | Visualization Quality | | | | Task Performance |
|---|---|---|---|---|---|
| | V-Acc.↑ | V-Red.↓ | V-Steps↑ | V-Ratio↑ | Accuracy↑ |
| MAZE | | | | | |
| MVoT with $\mathcal{L}_D$ | **0.9339** | **0.1010** | **8.4439** | **0.9449** | **0.9295** |
| MVoT without $\mathcal{L}_D$ | 0.6391 | 0.4931 | 5.6563 | 0.6930 | 0.7468 |
| MINIBEHAVIOR | | | | | |
| MVoT with $\mathcal{L}_D$ | **0.9681** | **0.0419** | **6.7532** | **0.9618** | **0.9514** |
| MVoT without $\mathcal{L}_D$ | 0.7939 | 0.2633 | 5.2333 | 0.7793 | 0.7228 |

- Visualization Correctness Ratio (V-Ratio): the average proportion of first k consecutive correct visualizations across the action sequence.

Due to the complexity of FROZENLAKE's image details, which makes automatic evaluation challenging, we only report quantitative results of visualizing the agent position in MAZE and MINIBEHAVIOR.

**Token discrepancy loss enhances accuracy and reduces redundancy in visualizations.** Table 3 shows that MVoT, enhanced with token discrepancy loss ($\mathcal{L}_D$), produces highly accurate visualizations with minimal pattern redundancy. Even in recursive generation scenarios, MVoT with $\mathcal{L}_D$ achieves an average of 95% correct and consecutive visualizations during reasoning. In contrast, the absence of $\mathcal{L}_D$ significantly degrades the generation quality: without $\mathcal{L}_D$, MVoT frequently generates redundant patterns and fails to accurately capture state transitions. These results align with findings from the image-editing scenario, as illustrated in Figure 10, which tracks quantitative metrics for MAZE at various epochs. Furthermore, poor visualization quality negatively impacts task performance, as highlighted in

the last column of Table 3, emphasizing the critical role of high-quality visualizations for better task outcomes. We also witness a performance drop on FROZENLAKE without token discrepancy loss from $0.8560$ to $0.7260$. Empirical evidence suggests that incorporating visual embedding with $\mathcal{L}_D$ in addition to $\mathcal{L}_C$ helps to bridge the gap between embeddings and improve the visual generation quality. This aligns with the findings by Tschannen et al. (2024) which noted similar limitations in embedding alignment for MLLMs.

## 7. Conclusion

We introduced Multimodal Visualization-of-Thought (MVoT), a novel reasoning framework that elicit reasoning process with multimodal thoughts using multimodal native generative models. MVoT outperformed textual reasoning baselines across a variety of tasks, meanwhile demonstrating better robustness to state complexity and offering enhanced interpretability. To ensure the generation of high-quality visualizations, we proposed the token discrepancy loss, addressing embedding misalignment in autoregressive MLLMs. This helps to alleviate the issues of redundant

patterns and inaccurate visual thought generation, leading to better task performance with MVoT. Furthermore, the complementary strengths of MVoT and Chain-of-Thought (CoT), as evidenced by their combined upper-bound performance, highlight the promise of hybrid multimodal reasoning approaches. This work underscores the value of multimodal cues and paves the way for future research into reasoning thoughts of hybrid modalities in complex tasks.

## Impact Statement

This paper aims to advance the field of multimodal reasoning and machine learning. While our work has potential societal implications, the specific datasets used in this study do not present any immediate concerns of this work that require explicit attention here.

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

# A. Additional Related Work

In addition to Section 2, we provide additional related work for the comprehensive understanding of this work.

**Video Generation World Model**    World models aim to predict the future for better understanding of the world, while video generation world models focus on the generation of videos to simulate and comprehend real-world dynamics (Zhu et al., 2024). Recent developments in generative models have markedly enhanced the capacity to capture physics and motion (Blattmann et al., 2023a;b; Esser et al., 2023; Zeng et al., 2024). The Sora model (Brooks et al., 2024) has demonstrated exceptional capabilities in this domain through its advanced generative techniques. World models extend beyond visual synthesis into numerous fields. In robotics, these models facilitate the simulation of physical interactions for diverse tasks (Zhen et al., 2024; Liu et al., 2024). They contribute to autonomous driving by predicting environmental changes (Wang et al., 2023; 2024b). Additionally, they function as effective simulators for data collection purposes (Yang et al., 2023a). Previous work typically generates outputs in a single modality. In contrast, our proposed MVoT introduces a novel reasoning method supporting multi-modal generation. This enhanced capability enables superior understanding, reasoning, and simulation of complex scenarios.

# B. Data

### B.1. Dataset Collection

MAZE dataset is generated using the `maze-dataset` framework (Ivanitskiy et al., 2023), leveraging an iterative depth-first search algorithm. Mazes of sizes 3, 4, 5, and 6 are created with varying random seeds. Navigation paths are generated, and we remove repetitive paths to prevent knowledge leakage between training and development sets. For each maze, potential destination candidates (three random coordinates) are selected alongside the true destination.

MINIBEHAVIOR dataset originates from the INSTALLINGAPRINTER simulation environment, where reinforcement learning (RL) agents are trained on grid environments of sizes 7 to 10 using `stable-baseline3` (Raffin et al., 2021). Action sequences that successfully complete the task are retained, ensuring no overlap with previously seen environments. For cases involving repeated action sequences or previously seen environments, the simulator layout is modified with specified probabilities: 40% chance of altering the printer or table coordinates respectively, 20% chance of randomly removing either the printer or table. The modified environments are re-evaluated using the previous action sequences to validate the outcomes.

FROZENLAKE dataset is based on Gym environments (Brockman, 2016) and scripts from Wu et al. (2024a). Trajectories are collected from RL agents selecting actions with maximum expected rewards using Q-table. Successful action sequences are retained if they are not seen in this environment. For unsuccessful sequences, cases where the agent falls into a hole have a 50% probability of being saved as-is or 50% with appended random actions. If the agent neither falls into a hole nor reaches the destination, the sequence is saved in accordance to Section 4. To increase the variance of the action sequences, we also randomly sample the paths when learning the Q-table following similar practice as above.

### B.2. Dataset Statistics

The statistics of the datasets are illustrated in Table 4. We also provide additional details about the datasets used in this work.

MAZE

- **Entities**: The starting point is marked with a red dot, with potential destination candidates labeled as A, B, C, and D.
- **Actions**: Go up, down, left, or right. All action sequences are valid, ensuring no collisions with walls and that every action is executable.
- **Visual Pattern**: Abstract sketch illustrating the maze layout and navigation path. The agent's path is visualized incrementally with red arrows, preserving all previous movements.

MINIBEHAVIOR

- **Entities**: The agent's current location represented by a red triangle, with a printer symbol and a brown-colored table.

*Table 4.* Statistics of the collected datasets, covering varying levels of complexity in actions and patterns.

| Task | MAZE | MINIBEHAVIOR | FROZENLAKE |
|---|---|---|---|
| Grid Sizes | 3-6 | 5-8 | 3-6 |
| Entity Types | 5 | 3 | 3 |
| Entities Numbers | 5 | 3 | 7.16 |
| Action Length | 9.11 | 7.83 | 6.56 |
| Action Types | 4 | 7 | 4 |
| Pattern Details | ✗ | ✗ | ✓ |
| Train Set Size | 5007 | 6400 | 6846 |
| Test Set Size | 1255 | 1604 | 1664 |

*Table 5.* Dataset statistical distributions of options.

| | | A | B | C | D | Total |
|---|---|---|---|---|---|---|
| MAZE | Train | 1187 | 1269 | 1305 | 1246 | 5007 |
| | Dev | 323 | 326 | 293 | 313 | 1255 |
| MINIBEHAVIOR | Train | 3321 | 1092 | 1456 | 531 | 6400 |
| | Dev | 834 | 297 | 349 | 124 | 1604 |
| FROZENLAKE | Train | 3043 | 2377 | 1426 | - | 6846 |
| | Dev | 735 | 580 | 349 | - | 1664 |

- **Actions**: Go up, down, left, or right; pick up; drop; toggle. The *pick up* action removes the printer from the map if next to the agent, while the *drop* action places the printer on the table if the agent is carrying it and next to the table.
- **Visual Pattern**: Abstract sketch of the environment layout. Unlike MAZE, only the agent's current position is visualized on the map, and whether the agent carries the printer is described in text.

**FROZENLAKE**

- **Entities**: The agent's current location depicted as an elf wearing a green hat, a gift and multiple holes on the frozen lake.
- **Actions**: Go up, down, left, or right.
- **Visual Pattern**: Comic-style illustrations with detailed depictions of the elf, background, and holes. Similar to MINIBEHAVIOR, the visualization is not incremental, focusing only on the current state.

Table 5 illustrates the distribution of correct choices of the train and dev split for each task. The distribution of different grid sizes in train and dev split for each task is listed in Table 6. Specifically, Table 7 shows how environmental complexity evolves with larger grid sizes in FROZENLAKE.

## C. Experiments

### C.1. Hyper-Parameters

Table 8 and 9 show the hyper-parameters for training MVoT and doing inference with GPT-4o.

All models were trained on MI300X GPUs. Table 8 provides the details of GPU configurations and hyperparameters for various experimental settings. For GPT-4o, we utilized the `2024-07-01` version hosted on the Azure platform, with inference parameters outlined in Table 9.

During MVoT training, as visual thoughts are recursively generated during inference, we applied input augmentation to improve visualization robustness and mitigate noise introduced by image reconstruction during tokenization and detokenization, as illustrated in Figure 4. This augmentation applies tokenization and detokenization over the input image for multiple times, with the iteration count randomly determined between 0 and 10.

*Table 6.* Dataset statistical distributions of grid sizes.

| MAZE | | | | |
|---|---|---|---|---|
| Grid Size | 3 | 4 | 5 | 6 |
| Train | 1481 | 1722 | 1823 | 1820 |
| Dev | 334 | 418 | 462 | 450 |
| MINIBEHAVIOR | | | | |
| Grid Size | 7 | 8 | 9 | 10 |
| Train | 1600 | 1600 | 1600 | 1600 |
| Dev | 401 | 401 | 401 | 401 |
| FROZENLAKE | | | | |
| Grid Size | 3 | 4 | 5 | 6 |
| Train | 308 | 1080 | 1397 | 2222 |
| Dev | 78 | 271 | 350 | 556 |

*Table 7.* Average number of key entities in FROZENLAKE with different grid sizes.

| Grid Size | 3 | 4 | 5 | 6 |
|---|---|---|---|---|
| Train | 4.7097 | 5.7166 | 7.4723 | 9.5049 |
| Dev | 4.6737 | 5.4689 | 7.4589 | 10.2267 |
| Total | 4.7030 | 5.6682 | 7.4696 | 9.648 |

*Table 8.* Hyper-parameters of fine-tuning Anole 7B for different system variants.

| Hyper-Parameters | *Direct* | CoT | *Interleaved* | MVoT |
|---|---|---|---|---|
| Random Seed | 42 | 42 | 42 | 42 |
| Epochs | 40 | 40 | 40 | 40 |
| Learning Rate | 0.0002 | 0.0002 | 0.0002 | 0.0002 |
| Train Batch Size | 4 | 4 | 4 | 4 |
| Val Batch Size | 16 | 16 | 8 | 8 |
| Grad Accumulation | 4 | 4 | 2 | 2 |
| GPUs | 8 | 8 | 32 | 32 |

*Table 9.* Hyper-parameters for GPT-4o

| | Temperature | Max Tokens | Top P | Frequency Penalty | Presence Penalty | Stop |
|---|---|---|---|---|---|---|
| GPT-4o | 0 | 800 | 1 | 0 | 0 | None |

## C.2. Prompting Templates

Table 10, 11, 12 shows examples of prompting templates and responses for each tasks with different system variants. Table 13 illustrate the prompting template for GPT-4o with MAZE and Table 14 shows an example for GPT-4o with MVoT on MINIBEHAVIOR. The other two tasks follow similar patterns.

## C.3. Reproducibility

We will release the code and the datasets at `URL-ANONYMOUS` upon acceptance for reproducibility purposes.

## D. Results

### D.1. Task Performance

Table 15 presents the detailed task performance metrics across varying grid sizes. For FROZENLAKE, we observe a noticeable decline in performance for both CoT and GPT-4o as the grid size increases, reflecting the growing complexity of the environment. This drop in performance highlights the limitations of these models in handling more intricate spatial reasoning tasks as the state complexity expands.

In contrast, MVoT demonstrates consistent performance across all grid sizes and tasks, underscoring its robustness to environmental complexity. Unlike CoT-based approaches, which struggle to generalize effectively in larger and more complex settings, MVoT's ability to integrate verbal and visual thoughts allows it to maintain stability and accuracy. The underlying intuition is that, even in more complex environments, MVoT focuses solely on the intended areas of the image without modifying irrelevant parts, thereby maintaining a consistent level of reasoning difficulty for the model. These results emphasize MVoT's resilience and adaptability, making it a better choice for tasks involving complex reasoning in dynamic environments.

### D.2. Visualizations

Table 16 presents fine-grained visualization metrics for MAZE and MINIBEHAVIOR across varying grid sizes. The consistent performance across grid sizes underscores the robustness of the visualizations. Compared to MVoT models trained without token discrepancy loss, those incorporating token discrepancy loss ($\mathcal{L}_D$) achieve better visualization accuracy and reduced redundancy, demonstrating the effectiveness of $\mathcal{L}_D$ in enhancing visualization quality.

Figures 6 and 7 provide examples of visualizations generated for MAZE and MINIBEHAVIOR. These examples clearly illustrate the improvement in visualization quality achieved by introducing token discrepancy loss.

Figure 10 illustrates the quantitative metrics for MAZE at various epochs in image-editing setting. With token discrepancy loss, the visualizations are more accurate and less redundant in the end, while converging faster as well.

**Image Tokenization in Autoregressive MLLMs** To understand why token discrepancy loss $\mathcal{L}_D$ helps to improve the quality of generated visual thoughts, we examine the two sets of embeddings introduced in autoregressive MLLMs: token embeddings for language modeling and visual embeddings for image tokenization. These embeddings originate from distinct systems because the visual codebook, image tokenizer, and detokenizer are trained separately from the autoregressive decoder, which creates a potential discrepancy between the two embedding spaces. To investigate their alignment, we compare the two embedding sets from Chameleon (Chameleon Team, 2024), LVM (Bai et al., 2024) and LlamaGen (Sun et al., 2024a) by calculating the average overlap ratio of the top-$k$ similar tokens based on cosine similarity. Results in Figure 5 reveal that, on average, only one token overlaps among the top 10 similar tokens, and approximately 20% of tokens overlap among the top 50 similar tokens. To further illustrate this discrepancy, we replace the image tokens with their most similar tokens in the token embeddings and visual embeddings, reconstructing the images (Figure 8 in Appendix D.2). After 10 iterations of tokenization and detokenization, the reconstructed images based on token embeddings exhibit significant differences from the originals, such as altered colors and distorted symbols (e.g., the

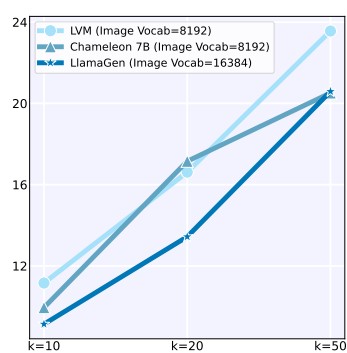

*Figure 5.* Average percentage of top-$k$ overlapping tokens.

*Table 10.* Example of input and output for Direct reasoning. *Italic* text is the expected response.

---

**Direct**

**MAZE**

Task: Maze Navigation Simulation
Determine the final destination (A, B, C or D) from the starting point (red point) following the action sequence. The definitions of the actions are as below.
* Go up/left/down/right: move one grid space in the absolute up/left/down/right direction.
Full Action Sequence: Go down. Go left. Go left. Go up. Go up.
Initial maze: <image>
Response: *The answer is A.*

**MINIBEHAVIOR**

Task: Mini-Behavior Installing the Printer
Determine whether the agent (red triangle) can pick up the printer (printer symbol) on the floor and place it on the table (brown area) and toggle it on. If not, identify the failure reason. The definitions of the actions are as below.
* Go up/left/down/right: move one grid space in the absolute up/left/down/right direction.
* Pick up: pick up the printer from the any of the grid next to the agent. If there is no printer next to the agent, the action fails.
* Drop: drop the printer to the table that is next to the agent. If there is no table next to the agent, the action fails.
* Toggle: toggle the printer that is on the table and next to the agent.
Return A, B, C or D.
Full Action Sequence: Go up. Go up. Go right. Go down. Go down. Pick up. Go right. Drop. Toggle.
A. Action Success.
B. Action Failed: Drop Error.
C. Action Failed: Pick Up Error.
D. Missing Key Objects.
Initial State: <image>
Response: *The answer is A.*

**FROZENLAKE**

Task: FrozenLake
Determine whether the agent (elf character) can safely reach the gift following the action sequence without falling into the holes. If not, identify the failure reason. The definitions of the actions are as below.
* Go up/left/down/right: move one grid space in the absolute up/left/down/right direction.
Return A, B or C.
Full Action Sequence: Go right. Go up.
A. Action Success.
B. Action Failed: Fall into the Hole.
C. Action Failed: Agent Safe but Fail to Reach Destination.
Initial State: <image>
Response: *The answer is A.*

*Table 11.* Example of input and output for CoT with environment layout. *Italic* text is the expected response.

---

**CoT with Environment Layout**

MAZE
Task: Maze Navigation Simulation
Determine the final destination (A, B, C or D) from the starting point (red point) following the action sequence. The definitions of the actions are as below.
* Go up/left/down/right: move one grid space in the absolute up/left/down/right direction.
Full Action Sequence: Go down. Go left. Go left. Go up. Go up.
Initial maze: <image>
Response: *Destination Coordinates: A Coordinate: [0, 0]. B Coordinate: [2, 0]. C Coordinate: [0, 1]. D Coordinate: [1, 1]. Initial Agent Coordinate: [2, 1]. Go down. Agent Coordinate: [2, 2]. Go left. Agent Coordinate: [1, 2]. Go left. Agent Coordinate: [0, 2]. Go up. Agent Coordinate: [0, 1]. Go up. Agent Coordinate: [0, 0]. Action sequence finished. The answer is A.*

MINIBEHAVIOR
Task: Mini-Behavior Installing the Printer
Determine whether the agent (red triangle) can pick up the printer (printer symbol) on the floor and place it on the table (brown area) and toggle it on. If not, identify the failure reason. The definitions of the actions are as below.
* Go up/left/down/right: move one grid space in the absolute up/left/down/right direction.
* Pick up: pick up the printer from the any of the grid next to the agent. If there is no printer next to the agent, the action fails.
* Drop: drop the printer to the table that is next to the agent. If there is no table next to the agent, the action fails.
* Toggle: toggle the printer that is on the table and next to the agent.
Return A, B, C or D.
Full Action Sequence: Go up. Go up. Go right. Go down. Go down. Pick up. Go right. Drop. Toggle.
A. Action Success.
B. Action Failed: Drop Error.
C. Action Failed: Pick Up Error.
D. Missing Key Objects.
Initial State: <image>Carrying objects: None.
Response: *Initial Agent Coordinate: [2, 7]. Initial Environment Layout: Printer Initial Coordinate: [3, 8]. Table Coordinates: [[5, 6], [5, 7], [6, 6], [6, 7], [7, 6], [7, 7]]. Go up. Agent Coordinate: [2, 6]. Carrying objects: None. Go up. Agent Coordinate: [2, 5]. Carrying objects: None. Go right. Agent Coordinate: [3, 5]. Carrying objects: None. Go down. Agent Coordinate: [3, 6]. Carrying objects: None. Go down. Agent Coordinate: [3, 7]. Carrying objects: None. Pick up. Agent Coordinate: [3, 7]. Carrying objects: printer_0. Go right. Agent Coordinate: [4, 7]. Carrying objects: printer_0. Drop. Agent Coordinate: [4, 7]. Carrying objects: None. Toggle. Agent Coordinate: [4, 7]. Carrying objects: None. Action sequence stopped. The answer is A.*

FROZENLAKE
Task: FrozenLake
Determine whether the agent (elf character) can safely reach the gift following the action sequence without falling into the holes. If not, identify the failure reason. The definitions of the actions are as below.
* Go up/left/down/right: move one grid space in the absolute up/left/down/right direction.
Return A, B or C.
Full Action Sequence: Go right. Go up.
A. Action Success.
B. Action Failed: Fall into the Hole.
C. Action Failed: Agent Safe but Fail to Reach Destination.
Initial State: <image>
Response: *Initial Agent Coordinate: [1, 2]. Initial Environment Layout: Holes Coordinate: [[1, 0], [0, 1]]. Gift Coordinate: [2, 1]. Go right. Agent Coordinate: [2, 2]. Go up. Agent Coordinate: [2, 1]. Action sequence stopped. The answer is A.*

*Table 12.* Example of input and output for CoT without environment layout. *Italic* text is the expected response.

---

**CoT without Environment Layout**

MAZE
Task: Maze Navigation Simulation
Determine the final destination (A, B, C or D) from the starting point (red point) following the action sequence. The definitions of the actions are as below.
* Go up/left/down/right: move one grid space in the absolute up/left/down/right direction.
Full Action Sequence: Go down. Go left. Go left. Go up. Go up.
Initial maze: <image>
Response: *Initial Agent Coordinate: [2, 1]. Go down. Agent Coordinate: [2, 2]. Go left. Agent Coordinate: [1, 2]. Go left. Agent Coordinate: [0, 2]. Go up. Agent Coordinate: [0, 1]. Go up. Agent Coordinate: [0, 0]. Action sequence finished. The answer is A.*

MINIBEHAVIOR
Task: Mini-Behavior Installing the Printer
Determine whether the agent (red triangle) can pick up the printer (printer symbol) on the floor and place it on the table (brown area) and toggle it on. If not, identify the failure reason. The definitions of the actions are as below.
* Go up/left/down/right: move one grid space in the absolute up/left/down/right direction.
* Pick up: pick up the printer from the any of the grid next to the agent. If there is no printer next to the agent, the action fails.
* Drop: drop the printer to the table that is next to the agent. If there is no table next to the agent, the action fails.
* Toggle: toggle the printer that is on the table and next to the agent.
Return A, B, C or D.
Full Action Sequence: Go up. Go up. Go right. Go down. Go down. Pick up. Go right. Drop. Toggle.
A. Action Success.
B. Action Failed: Drop Error.
C. Action Failed: Pick Up Error.
D. Missing Key Objects.
Initial State: <image>Carrying objects: None.
Response: *Initial Agent Coordinate: [2, 7]. Go up. Agent Coordinate: [2, 6]. Carrying objects: None. Go up. Agent Coordinate: [2, 5]. Carrying objects: None. Go right. Agent Coordinate: [3, 5]. Carrying objects: None. Go down. Agent Coordinate: [3, 6]. Carrying objects: None. Go down. Agent Coordinate: [3, 7]. Carrying objects: None. Pick up. Agent Coordinate: [3, 7]. Carrying objects: printer_0. Go right. Agent Coordinate: [4, 7]. Carrying objects: printer_0. Drop. Agent Coordinate: [4, 7]. Carrying objects: None. Toggle. Agent Coordinate: [4, 7]. Carrying objects: None. Action sequence stopped. The answer is A.*

FROZENLAKE
Task: FrozenLake
Determine whether the agent (elf character) can safely reach the gift following the action sequence without falling into the holes. If not, identify the failure reason. The definitions of the actions are as below.
* Go up/left/down/right: move one grid space in the absolute up/left/down/right direction.
Return A, B or C.
Full Action Sequence: Go right. Go up.
A. Action Success.
B. Action Failed: Fall into the Hole.
C. Action Failed: Agent Safe but Fail to Reach Destination.
Initial State: <image>
Response: *Initial Agent Coordinate: [1, 2]. Go right. Agent Coordinate: [2, 2]. Go up. Agent Coordinate: [2, 1]. Action sequence stopped. The answer is A.*

*Table 13.* Prompting template for GPT-4o zero-shot direct and CoT inference, with MAZE as example.

---

**GPT-4o on MAZE**

**GPT-4o zero-shot inference**

Task: Maze Navigation Simulation
Determine the final destination (A, B, C or D) from the starting point (red point) following the action sequence. The definitions of the actions are as below.
* Go up/left/down/right: move one grid space in the absolute up/left/down/right direction.
Full Action Sequence: Go down. Go left. Go left. Go up. Go up.
Initial maze: <image>
Conclude your final answer between <ANSWER>and </ANSWER >.

- - - - - - - - - - - - - - - - - - - - - - - - - - - - - - - - - - - - - - - - - - - - - - - - - - -

**GPT-4o zero-shot CoT**

Task: Maze Navigation Simulation
Determine the final destination (A, B, C or D) from the starting point (red point) following the action sequence. The definitions of the actions are as below.
* Go up/left/down/right: move one grid space in the absolute up/left/down/right direction.
Full Action Sequence: Go down. Go left. Go left. Go up. Go up.
Initial maze: <image>
Let's think step by step. Conclude your final answer between <ANSWER>and </ANSWER >.

---

elf's appearance). This contrast highlights the misalignment between the two embedding systems. By introducing token discrepancy loss, we hope to bridge this gap, enabling better alignment between visual and token embeddings, thus allowing the model to generate higher-quality visual rationale.

Figure 9 provides an example where CoT fails on FROZENLAKE while MVoT succeeds. This example clearly and intuitively demonstrates that CoT is highly sensitive to environmental complexity and generates incorrect coordinate descriptions of holes when dealing with FROZENLAKE. In contrast, MVoT successfully avoids this issue by leveraging the visualization of thought during reasoning.

## E. Limitation

MVoT unifies the verbal and visual thoughts to elicit the reasoning process through image visualizations. However, we observe that the generated visualizations often attempt to reconstruct task-irrelevant details, such as the background patterns in FROZENLAKE, while overlooking the intended alterations in the visualizations. This can be mitigated by incorporating the guidance techniques as in image generation with diffusion models (Sadat et al., 2024), which we propose as a direction for future improvement. Additionally, explicitly generating visualizations introduces computational overhead during inference. To address this, we advocate for further research into compact image representations using fewer tokens (Choudhury et al., 2024; Yu et al., 2024), thereby reducing the computational cost of visualization generation.

*Table 14.* Prompting template for GPT-4o with MVoT, with MINIBEHAVIOR as example. <VISUALIZATION> is generated by MVoT as the plug-in. *Italic* text is the response generated by GPT-4o.

---

**GPT-4o with MVoT on MINIBEHAVIOR**

To predict the consequence of a sequence of actions, you need to simulate the action sequence with 'one action' at a time. Each time after simulating one action, 'your turn is finished' and you should wait for the environment to provide the corresponding image observation.
When all the actions are completed or a target situation is achieved, conclude the final answer.
The interaction between you and the environment at step i is as following:
Your turn:
First conduct reasoning to determine to simulate 'action' or to conclude the 'answer' based on the previous observation 'Observation {i-1}'.
Reasoning {i}: {your thinking before simulating 'action' or concluding 'answer' if all actions are completed or a target situation aligned with one of the options is achieved. }

If to simulate the next action, return as follows.
Reasoning {i}: {your thinking before simulating 'action i' based on the full action sequence}
Action {i}: {i-th action to simulate} # from the full action sequence

If to conclude the answer based on the observation that aligns with one of the options, return as follows.
Reasoning {i}: {answer can be concluded or actions are completed based on the observation i-1. }
Answer: The answer is {Choose from multiple-choice options}

In reasoning, don't describe the image, only determine whether to conclude the answer if previous observation aligns with one of the options (pay attention to the key objects) or to simulate the next action.

Don't mention misleading word such as safe or anything will cause hurt.

Task: Mini-Behavior Installing the Printer
Determine whether the agent (red triangle) can pick up the printer (printer symbol) on the floor and place it on the table (brown area) and toggle it on. If not, identify the failure reason. The definitions of the actions are as below.
* Go up/left/down/right: move one grid space in the absolute up/left/down/right direction.
* Pick up: pick up the printer from the any of the grid next to the agent. If there is no printer next to the agent, the action fails.
* Drop: drop the printer to the table that is next to the agent. If there is no table next to the agent, the action fails.
* Toggle: toggle the printer that is on the table and next to the agent.
Return A, B, C or D.
Full Action Sequence: Go right. Pick up. Go up. Drop. Toggle.
A. Action Success.
B. Action Failed: Drop Error.
C. Action Failed: Pick Up Error.
D. Missing Key Objects.
<image>
Conduct reasoning first to determine whether 1) to simulate the action sequence with 'one step' at a time in the format of 'Step'/'Conclusion', 'Reasoning' and 'Action', or 2) to onclude the answer when 'previous observation aligns with one of the options' or 'all the actions are completed'. Do not generate the observation by yourself.

*Reasoning 1: The agent (red triangle) is currently positioned next to the printer. The first action in the sequence is to move right. I will simulate this action to see if the agent can move closer to the printer.*
*Action 1: right.*
*Observation 1: <VISUALIZATION>*

*Table 15.* Fine-grained task performance on different grid sizes

| MAZE | | | | | |
|---|---|---|---|---|---|
| Grid size | 3 | 4 | 5 | 6 | Overall |
| GPT-4o Direct | 0.6667 | 0.7196 | 0.7457 | 0.6888 | 0.7100 |
| GPT-4o CoT | 0.9103 | 0.7639 | 0.7457 | 0.6978 | 0.7386 |
| *Direct* | 0.5385 | 0.7528 | 0.7457 | 0.7067 | 0.7171 |
| CoT (w env layout) | 1.0000 | 0.9926 | 0.9800 | 0.9690 | 0.9792 |
| *Interleaved* | 0.9487 | 0.9188 | 0.8857 | 0.8197 | 0.8677 |
| MVoT | 0.9103 | 0.9668 | 0.9257 | 0.9162 | 0.9295 |
| MINIBEHAVIOR | | | | | |
| Grid size | 7 | 8 | 9 | 10 | Overall |
| GPT-4o Direct | 0.3965 | 0.4389 | 0.4838 | 0.5112 | 0.4576 |
| GPT-4o CoT | 0.4663 | 0.4439 | 0.4788 | 0.4813 | 0.4676 |
| *Direct* | 0.6733 | 0.7207 | 0.7758 | 0.7307 | 0.7250 |
| CoT (w env layout) | 0.9800 | 0.9825 | 0.9723 | 0.9900 | 0.9813 |
| *Interleaved* | 0.8279 | 0.8329 | 0.8615 | 0.8404 | 0.8406 |
| MVoT | 0.9651 | 0.9676 | 0.9528 | 0.9202 | 0.9514 |
| FROZENLAKE | | | | | |
| Grid size | 3 | 4 | 5 | 6 | Overall |
| GPT-4o Direct | 0.6916 | 0.5215 | 0.4372 | 0.3933 | 0.4976 |
| GPT-4o CoT | 0.6018 | 0.5000 | 0.4113 | 0.3911 | 0.4664 |
| *Direct* | 0.8263 | 0.8038 | 0.7468 | 0.7533 | 0.7788 |
| CoT (w env layout) | 0.9401 | 0.7225 | 0.5000 | 0.3911 | 0.6148 |
| *Interleaved* | 0.7305 | 0.6818 | 0.6017 | 0.5956 | 0.6460 |
| MVoT | 0.8623 | 0.8397 | 0.8377 | 0.8876 | 0.8560 |

*Table 16.* Fine-grained task performance and visualization metrics on different grid sizes.

| | MAZE | | | | | | | |
|---|---|---|---|---|---|---|---|---|
| | 3 | | 4 | | 5 | | 6 | |
| | MVoT w/ $\mathcal{L}_D$ | MVoT w/o $\mathcal{L}_D$ | MVoT w/ $\mathcal{L}_D$ | MVoT w/o $\mathcal{L}_D$ | MVoT w/ $\mathcal{L}_D$ | MVoT w/o $\mathcal{L}_D$ | MVoT w/ $\mathcal{L}_D$ | MVoT w/o $\mathcal{L}_D$ |
| Task Acc | 0.9103 | 0.7949 | 0.9668 | 0.7970 | 0.9257 | 0.7657 | 0.9162 | 0.7031 |
| V-Acc. | 0.9654 | 0.8447 | 0.9719 | 0.7858 | 0.9549 | 0.6786 | 0.9094 | 0.5609 |
| V-Red. | 0.0769 | 0.1923 | 0.0553 | 0.3284 | 0.0829 | 0.4200 | 0.1275 | 0.5993 |
| V-Steps | 4.6538 | 4.0513 | 6.8930 | 5.4576 | 7.9771 | 5.5943 | 10.0346 | 6.0219 |
| V-Ratio | 0.9489 | 0.8314 | 0.9677 | 0.7897 | 0.9545 | 0.7195 | 0.9264 | 0.6087 |
| | MINIBEHAVIOR | | | | | | | |
| | 7 | | 8 | | 9 | | 10 | |
| | MVoT w/ $\mathcal{L}_D$ | MVoT w/o $\mathcal{L}_D$ | MVoT w/ $\mathcal{L}_D$ | MVoT w/o $\mathcal{L}_D$ | MVoT w/ $\mathcal{L}_D$ | MVoT w/o $\mathcal{L}_D$ | MVoT w/ $\mathcal{L}_D$ | MVoT w/o $\mathcal{L}_D$ |
| Task Acc | 0.9651 | 0.7606 | 0.9676 | 0.7980 | 0.9528 | 0.6850 | 0.9202 | 0.6425 |
| V-Acc. | 0.9781 | 0.7960 | 0.9694 | 0.8525 | 0.9791 | 0.7928 | 0.9479 | 0.7497 |
| V-Red. | 0.0349 | 0.2668 | 0.0399 | 0.1696 | 0.0367 | 0.2651 | 0.0898 | 0.3290 |
| V-Steps | 5.6708 | 4.3666 | 6.1147 | 5.0374 | 7.6640 | 5.6089 | 8.6259 | 5.9663 |
| V-Ratio | 0.9693 | 0.7791 | 0.9608 | 0.8350 | 0.9786 | 0.7752 | 0.9313 | 0.7258 |

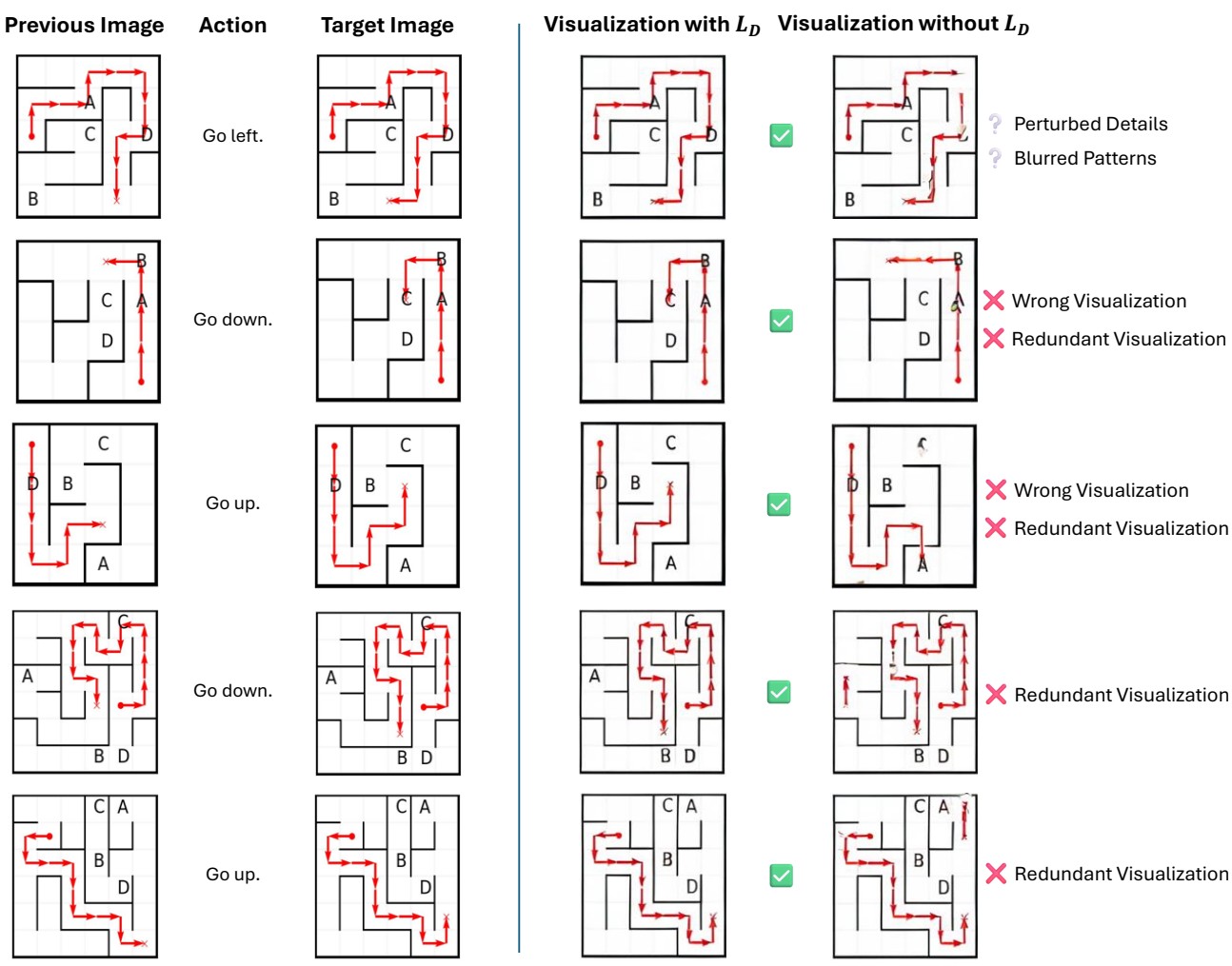

*Figure 6.* Qualitative analysis of MAZE visualization quality from systems trained with and without token discrepancy loss ($\mathcal{L}_D$) in image-editing setting.

| Previous Image | Action | Target Image | Visualization with $L_D$ | Visualization without $L_D$ |
|---|---|---|---|---|
| | Go down. | | ✅ | ❓ Perturbed Details |
| | Pick up. | | ✅ | ❌ Wrong Visualization
Pick up action failed. |
| | Go down. | | ✅ | ❌ Wrong Visualization
❌ Redundant Visualization |
| | Go right. | | ✅ | ❌ Redundant Visualization |
| | Go left. | | ✅ | ❌ Redundant Visualization |

*Figure 7.* Qualitative analysis of MINIBEHAVIOR visualization quality from systems trained with and without token discrepancy loss ($\mathcal{L}_D$) in image-editing setting.

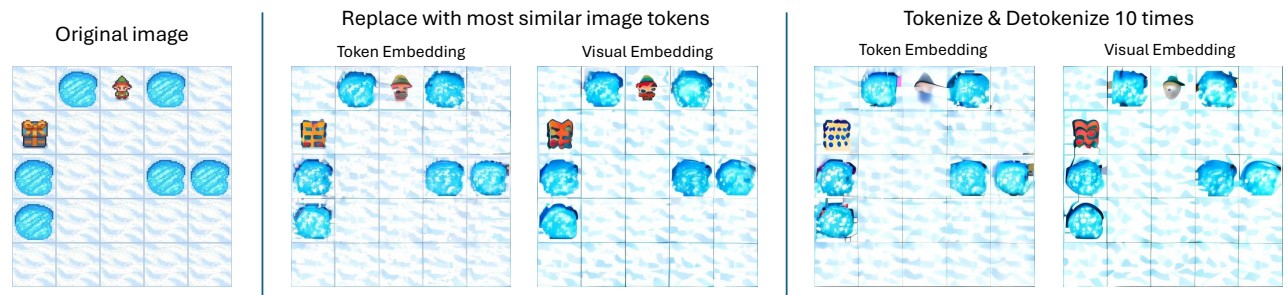

*Figure 8.* Reconstruct the image with Anole by 1) replacing the image tokens with most similar tokens in BPE and visual embeddings; 2) tokenizing and detokenizing the image after replacement for 10 times.

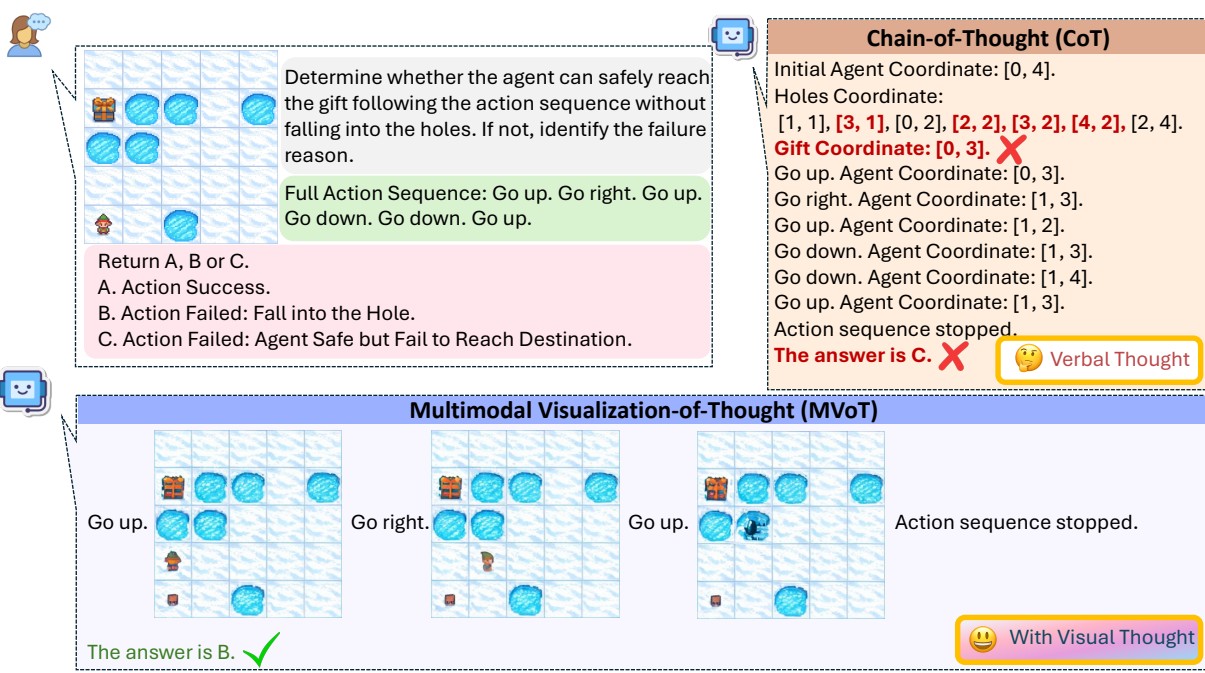

*Figure 9.* An example where CoT fails on FROZENLAKE while MVoT succeeds. Because of sensitivity to the environment, CoT generated incorrect coordinate descriptions of holes, leading to the wrong answer.

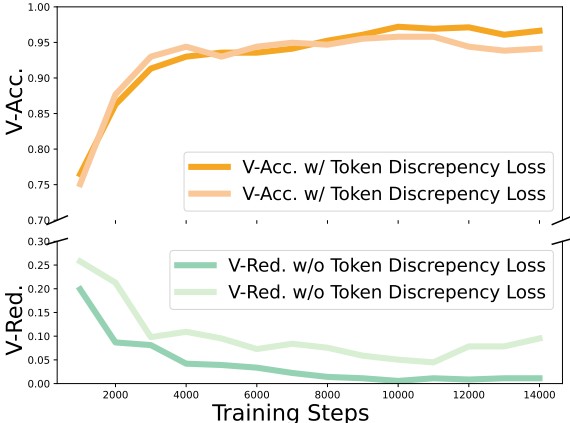

*Figure 10.* Visualization metrics for image editing on MAZE, evaluated using 800 randomly sampled examples.

