# OpenReview forum: "Imagine While Reasoning in Space: Multimodal Visualization-of-Thought"
_ICML.cc/2025/Conference — ICML 2025 poster_

### Official Review · Reviewer_2g3q · 2025-02-18

**Overall Recommendation:** 3

**Summary:**

The paper introduces Multimodal Visualization-of-Thought (MVoT), a new reasoning paradigm designed to enhance the spatial reasoning capabilities of Multimodal Large Language Models. It improves the spatial reasoning ability over Chain-of-Thought (CoT) prompting by generating image visualizations of their reasoning traces, effectively allowing them to "think" in both words and images. Experiments on three datasets demonstrate the effectiveness of the proposed method.

**Claims And Evidence:**

### Well-supported claims
1. MVoT outperforms CoT in certain complex spatial reasoning tasks.
2. MVoT provides better interpretability than CoT.

### Weakly Supported Claims
MVoT generalizes better than CoT: the paper does not sufficiently test out-of-domain generalization beyond the three controlled, grid-based environments. For example, robotics, navigation, and real-world images are not evaluated.

**Essential References Not Discussed:**

To my knowledge, necessary references are discussed.

**Experimental Designs Or Analyses:**

- The three benchmarks (MAZE, MINIBEHAVIOR, FROZENLAKE) gradually increase in complexity, testing different aspects of spatial reasoning.
- Ablation studies on token discrepancy loss show that removing LD worsens visual coherence and reasoning accuracy, validating its necessity.
- The comparison against several baselines (Direct Prompting, CoT, GPT-4o, and Interleaved Training) demonstrate its strengths.
- Beyond task accuracy dominates, the paper also introduces visualization accuracy, redundancy, and pattern correctness metrics, providing some insight into how well the model generates visual thought.

**Methods And Evaluation Criteria:**

The proposed method and evaluation criteria make sense for the problem. Please also see weaknesses below.

**Other Comments Or Suggestions:**

NA

**Other Strengths And Weaknesses:**

### Strengths
- It extends CoT to multimodal reasoning by integrating visual thought generation. First work to natively interleave image and text reasoning in MLLMs.
- This paper introduces token discrepancy loss, which helps align text and image token spaces, improving coherence and quality of generated visuals.
- The performance improvement on three benchmarks demonstrate the effectiveness of the proposed method.

### Weaknesses
- All tasks (Maze, MiniBehavior, FrozenLake) are toy grid-worlds based experiments. No experiments on real-world benchmarks such as images, 3D reasoning, or robotics.
- The model has been fine-tuned on the train set of the benchmark. No zero-shot or out-of-distribution evaluations. It’s unclear if MVoT can handle new spatial reasoning problems beyond its training setup.

**Questions For Authors:**

- Can the authors discuss some specific failure cases of the proposed method?
- Is there a specific reason for using Chameleon as the base model? Can the proposed method work for other unified vision-language models?

**Relation To Broader Scientific Literature:**

This paper utilizes/fine-tunes a unified multimodal model Chameleon to enable interleaved multimodal output for chain-of-thought reasoning. It extends the textual CoT to the multimodal area.

**Theoretical Claims:**

No theoretical claim.

---

> ### Author Rebuttal · Authors · 2025-04-01
>
> Thank you for your recognition of our work and your valuable suggestions. We would like to address your comments as follows to get more support:
>
> **Toy grid-world based experiments**
>
> Our use of grid-based benchmarks offers better controllability and systematic investigation across various aspects of spatial reasoning—from pattern complexity to action complexity. Grid-based benchmarks also enable easier evaluation towards whether the generated visualization is correct in terms of the spatial transition between steps rather than focusing on the image pixel details.
>
> We agree that it would be interesting to see how MVoT performs on real-world reasoning tasks. However, due to the current lack of interleaved text-image reasoning datasets, it’s hard for us to adapt MVoT to real-world scenarios such as robotics or 3D reasoning in this work. But we hope our paper as the first exploration of natively generating multimodal reasoning traces offers foundational insights and inspires further studies in this direction.
>
> **Zero-shot / out-of-distribution evaluations**
>
> We appreciate your suggestion to evaluate MVoT in cross-task generalization settings. The inter-task OOD performance of MVoT relies on image generation ability. Currently, due to computational constraints, we fine-tune the model using LoRA on a limited dataset, which restricts its ability and makes it more challenging to generalize across tasks and scenarios.
>
> However, we conducted preliminary experiments to test MVoT’s ability to generalize to OOD grid sizes. We found that the model can successfully generate up to 7 consecutive visualizations on larger grids, adapting its stepwise spatial transitions accordingly. That said, we also observed occasional redundancies and inaccuracies, indicating room for improvement.
>
> We plan to explore MVoT’s OOD behavior more extensively as future work, and will include further discussions on this topic in the camera-ready version, should the paper be accepted.
>
> **Failure cases of the proposed method**
>
> Most of the failure cases for MVoT are caused by generating incorrect visualizations during inference, including generating wrong or redundant visualizations with perturbed or blurred image details, as we illustrated in Section 6 and Appendix D.2. We will include  more analysis and discussion on failure cases in our camera-ready version upon acceptance.
>
> **Model choices**
>
> Chameleon supports interleaved text-image generation, which meets the requirements of MVoT. In principle, MVoT can be extended to other multimodal models that support interleaved text-image generation. However, many current MLLMs are restricted to producing textual outputs only. We look forward to the development of architectures that support richer, interleaved multimodal outputs, which would further broaden the applicability of MVoT.
>
> We thank the reviewer for appreciating the novelty and the constructive feedback on our work. We will reflect these considerations in our camera-ready submission upon acceptance.

---

> > ### Comment · Reviewer_2g3q · 2025-04-01
> >
> > Thank you for the responses. Most of my concerns are addressed. \
> > Regarding model choices, I think there are currently more and more unified models that can generate both images and texts like Janus, VILA-U, etc. It would be interesting to know if the proposed method works for them as well. Overall the selection of the model is limited to (a fine-tuned) Chameleon, but it's a promising line of work - so I recommend acceptance of this paper.

---

> > > ### Author Response · Authors · 2025-04-02
> > >
> > > Thank you for your kind response. Regarding model choices, we are equally excited by the recent emergence of unified models capable of both text and image generation, such as Janus and VILA-U. However, it is important to note that not all unified models are designed for **interleaved modal** generation. For example, both Janus and VILA-U adopt a **mixed-modality** training paradigm, meaning they are trained on paired modalities—such as [image, text], [text, image], or etc.—but generate output conditioned on the former modality in each pair. As noted in the VILA-U paper: "*We use [image, text], [text, image], and [text, video] forms, with supervision loss added only on the latter modality in each pair to avoid unconditional content generation and promote modality alignment.*"
> > >
> > > This training strategy, while effective for paired generation tasks, does not naturally align with the interleaved modal generation setting that MVoT is designed for—where sequences of texts and images are generated in tandem. In contrast, Chameleon supports interleaved modal generation during pre-training, making it a suitable and practical choice for our initial exploration of the MVoT strategy.
> > >
> > > We fully agree that applying MVoT to a broader range of models and task settings is a promising and interesting direction. We are currently working on this and hope to work out a more general solution in the future. We hope our work offers foundational insights and inspires further studies in this direction.
> > >
> > > Once again, we sincerely appreciate your time and constructive comments. If our responses have addressed your concerns, we hope you may consider raising your score.

---

### Official Review · Reviewer_wVwN · 2025-03-13

**Overall Recommendation:** 3

**Summary:**

This paper proposes a new multimodal reasoning paradigm — Multimodal Visualization-of-Thought (MVoT), which enables the model to "think" in both textual and visual spaces interleaved. The authors implement this by fine-tuning a Chameleon-like model, Anole-7B, to generate interleaved text and images. They collect data and fine-tune the model on three spatial reasoning tasks. Extensive experiments show that MVoT benefits spatial reasoning, outperforming CoT and direct prompting. The generated visual thoughts can also improve closed-source models.

**Claims And Evidence:**

The claims in this paper are supported either by references or by experiment results.
The experiment results are comprehensive and convictive.

**Essential References Not Discussed:**

I acknowledge the discussion of spatial reasoning papers like SpatialVLM and SpatialRGPT in this paper. It would be better to discuss some essential recent multimodal spatial reasoning fashions, for example:
[A] Does Spatial Cognition Emerge in Frontier Models?
[B] Thinking in Space: How Multimodal Large Language Models See, Remember, and Recall Spaces

**Experimental Designs Or Analyses:**

I have reviewed the experimental design and analysis of the results.

**Methods And Evaluation Criteria:**

The proposed fine-tuned Anole implementation is reasonable and clever for the task. The used benchmarks and metrics are technically sound for evaluating complex spatial reasoning abilities.

**Other Comments Or Suggestions:**

Generally, I appreciate the idea of this paper. However, I think there are some missing pieces that can probably push this work to a better quality. I would like to raise my score if the author can address my concerns during the rebuttal, but I might also turn it down if the rebuttal is weak.

**Other Strengths And Weaknesses:**

**Strengths**
The proposed new paradigm enables reasoning beyond text space, unlocking more potential abilities and applications of unified MLLMs. The idea is simple but intuitive.

**Weakness**
1. As there is no constraint that the text (action) can be consistent with visualization, there is a way that the visualization can make the
reasoning process even more vulnerable. As shown in the analysis in Fig. 4, the visualization can modify the background or just not align with the action. What's worse, I think the image prediction is more vulnerable than text prediction, so the error accumulation in a long reasoning chain can be huge.

2. All shown examples and experimental results are in-distribution. It would be more interesting to the generalization between different spatial reasoning tasks. For example, MLLM learns on the MAZE task, and evaluates on MINIBEHAVIOR and FROZENLAKE or other MAZE games. I suspect such MVoT would be vulnerable in OOD cases, since predicting visualization is easier to accumulate errors during reasoning, while all existing experimental results are all in-distribution. Besides, in the practice cases, the MLLMs always meet up with the OOD cases instead of in-distribution cases. I would like to see such a comparison between MVoT vs Direct and CoT.

**Questions For Authors:**

This paper reminds me of [C], which generates image visualizations to help robot control. The scenarios shown in this paper are mainly game engines, but it would be interesting to see whether this can work on more realistic spatial reasoning tasks, like SpatialVLM, SpatialRGPT and [B].

[B] Thinking in Space: How Multimodal Large Language Models See, Remember, and Recall Spaces
[C] Learning Universal Policies via Text-Guided Video Generation

**Relation To Broader Scientific Literature:**

The paper provided a promising solution for multimodal reasoning and a great motivation to develop the unified multimodal model.

**Theoretical Claims:**

I have checked the correctness of the formula.

---

> ### Author Rebuttal · Authors · 2025-04-01
>
> Thank you for your recognition of our work and your valuable suggestions. We would like to address your comments as follows to get more support:
>
> **Visualization Consistency and Vulnerability**
>
> We acknowledge the concern that unconstrained visualization could introduce inconsistencies, particularly if visual outputs do not precisely match the underlying reasoning steps. However, in our framework, visualization is not a standalone prediction but is tightly coupled with the reasoning chain. The generated visualization is conditioned on both the current reasoning step and the accumulated context. This is similar to textual rationales, which also lack hard constraints on internal consistency but are still useful in guiding reasoning. Importantly, MVoT does not generate full images from scratch; instead, it predicts spatial transitions between steps, which is more structured and less generative-intensive. This design significantly reduces the likelihood of arbitrary or unaligned visual outputs.
>
> From a performance perspective, while pure text-based reasoning can be effective in general tasks, prior work [1, 2] and our own experiments on FrozenLake show that it falls short in complex multimodal spatial reasoning. In contrast, MVoT demonstrates more robust performance by leveraging visual signals, suggesting that visualization acts as a regularizer rather than a vulnerability.
>
> Lastly, we would like to clarify that concerns around the generalization ability are derived from the image generation model, which does not necessarily represent the drawback of MVoT as a reasoning method.
>
> **Error Accumulation in Visual Prediction**
>
> We agree that cascading errors are a valid concern in any multi-step reasoning framework, especially in modalities like vision. However, it’s important to note that text-only reasoning is equally susceptible to error accumulation, especially when dealing with ambiguous or spatially grounded tasks. For example, our FrozenLake results show that when errors occur in describing environment layouts, Chain-of-Thought can even underperform the Direct baseline due to compounding misrepresentations.
>
> In contrast, our results show that MVoT performs comparably or better than both Direct and CoT approaches, suggesting that the visual modality does not necessarily amplify errors, and may in fact help mitigate them by providing an interpretable, stepwise grounding of spatial transitions.
>
> **Out-of-Distribution (OOD) Generalization**
>
> We appreciate your suggestion to evaluate MVoT in cross-task generalization settings. This is an important direction. The inter-task OOD performance of MVoT relies on image generation ability across scenarios. Currently, due to computational constraints, we fine-tune the model using LoRA on a limited dataset, which restricts its ability and makes it more challenging to generalize across tasks such as MAZE to MINIBEHAVIOR.
>
> However, we conducted preliminary experiments to test MVoT’s ability to generalize to OOD grid sizes. We found that the model can successfully generate up to 7 consecutive visualizations on larger grids, adapting its stepwise spatial transitions accordingly. That said, we also observed occasional redundancies and inaccuracies, indicating room for improvement.
>
> We plan to explore MVoT’s OOD behavior more extensively as future work, and will include further discussions on this topic in the camera-ready version, should the paper be accepted.
>
> **More realistic spatial reasoning tasks**
>
> We agree that it would be interesting to see how MVoT performs on realistic spatial reasoning tasks. However, due to the current lack of interleaved text-image reasoning datasets, it’s hard for us to adapt MVoT to these scenarios in this work. We believe our paper—being one of the first to explore native multimodal reasoning trace generation—lays important groundwork for future research, and we hope it inspires further development of models and datasets in this space.
>
> We thank the reviewer for providing us with more recent references, which we would include in our discussion together with the comments above in our camera-ready version upon acceptance.
>
> ```
> Reference
> [1] Ramakrishnan, Santhosh Kumar, et al. "Does Spatial Cognition Emerge in Frontier Models?." arXiv preprint arXiv:2410.06468 (2024).
> [2] Wang, Jiayu et al. “Is A Picture Worth A Thousand Words? Delving Into Spatial Reasoning for Vision Language Models.” The Thirty-eighth Annual Conference on Neural Information Processing Systems, 2024.
> ```

---

### Official Review · Reviewer_yEfK · 2025-03-14

**Overall Recommendation:** 3

**Summary:**

This paper presents Multimodal Visualization-of-Thought (MVoT). This paradigm enables visual thinking in MLLMs by generating image visualizations of their reasoning traces. MVoT is motivated by human's cognition, having the ability to think both in words and images seamlessly.

MVoT is developed based on Chameleon model, which natively unifies the text and image generation and understanding.

In MVoT, the model is trained to output a visualization after every intermediate verbal step, and a token descrepancy loss is introduced to enhance the quality of generated image.

In experiments, the author develop MVoT based on Anole-7B, a finetuned version of Chameleon model and train the model on three tasks.

Through experiments, the author shows the effectiveness of MVoT, by comparing the model with different finetuning strageties as well as GPT-4o.

**Claims And Evidence:**

The authors claim that MVoT is competitive with other tasks, and more rubust and reliable than CoT finetuning.
Thought experiments in Section 5, the author support these claims by a clear set of experiments.

**Essential References Not Discussed:**

No

**Experimental Designs Or Analyses:**

Yes, please refer to paper summary.

**Methods And Evaluation Criteria:**

The authors use three proper public benchmarks to train and test the MVoT method, which makes sense.

**Other Comments Or Suggestions:**

No

**Other Strengths And Weaknesses:**

Strengths:
1. This paper propose a new multimodal reasoning paradigm, which is much more intuitive than traditional methods especially on multimodal interactive tasks.
2. The experiment is well designed and many insights are provided when developing this method.

Weakness:
1. The experiment scope is narrow. Only three tasks are included, and the environments are not complex enough, with fixed and small set of actions and objects. So it is unclear whether such method can be effectively applied to the real world reasoning tasks.

**Questions For Authors:**

1. How many visual tokens are use in each visualization? In Chameleon, each image is generated with 1024 tokens, I assume less number of tokens are used in each visualization? Based on this question, if this method is applied to more complex environment, do we need more tokens? I assume much more tokens are needed in more complex setting, if this is the case, the reasoning could be very inefficient (e.g. 1024 tokens for each step).
2. I'm not very sure about the intuition of Token Discrepancy Loss. Is it encouraging the visual tokens to be more close to each other, or is it discouraging the model to output the visual tokens that deviate from the global embedding distribution?

**Relation To Broader Scientific Literature:**

The method proposed by this paper is relevant to many new multimodal benchmarks, like EMMA [1]. And many other application such as robotics, interactive image editing.
It is potentially a new paradigm of multimodal reasoning, so the contribution is relevant to any benchmark or multimodal models.


[1] Hao, Y., Gu, J., Wang, H. W., Li, L., Yang, Z., Wang, L., & Cheng, Y. (2025). Can MLLMs Reason in Multimodality? EMMA: An Enhanced MultiModal ReAsoning Benchmark. arXiv preprint arXiv:2501.05444.

**Theoretical Claims:**

There is no complex proofs for theoretical claim that need to be checked in this paper, it is an empirical study and all proposed methods are justified by experiments.

---

> ### Author Rebuttal · Authors · 2025-04-01
>
> Thank you for your recognition of our work and your valuable suggestions. We would like to address your comments as follows to get more support:
>
> **Experiment and task scope**
>
> Our use of grid-based benchmarks offers better controllability and systematic investigation across various aspects of spatial reasoning—from pattern complexity to action complexity. Grid-based benchmarks also enable easier evaluation towards whether the generated visualization is correct in terms of the spatial transition between steps rather than focusing on the image pixel details.
>
> We agree that it would be interesting to see how MVoT performs on real-world reasoning tasks. However, due to the current lack of interleaved text-image reasoning datasets, it’s hard for us to adapt MVoT to real-world scenarios in this work. But we hope our paper as the first exploration of natively generating multimodal reasoning traces offers foundational insights and inspires further studies in this direction.
>
> **Visual tokens**
>
> We generate 1024 visual tokens per visualization. Bounded by a 4096-token context limit, to manage this efficiently, when implementation, we employ a recursive generation strategy with a Markovian assumption: the visualization of the next step $v_{i+1}$ is derived based on the visualizations of the previous step $v_{i}$ and the initial step $v_{0}$. This assumption holds across all the benchmarks we used in this work since at each step the information of the visualization (with textual description for MiniBehavior) is complete.
>
> While this design is well-suited to current benchmarks, we acknowledge the opportunity to improve scalability—for instance, through more compact visual representations, as we stated in Appendix E Limitation. As the first work to explore generating native multimodal reasoning traces, MVoT opens up new possibilities, and we hope it will serve as a foundation for future advancements in more complex multimodal tasks.
>
> **Intuition of token discrepancy loss**
>
> Token discrepancy loss discourages the model to output visual tokens that deviate too much from the corresponding golden visual tokens. By penalising deviations from ground-truth visual tokens, it effectively guides the model toward more faithful and visually semantically aligned generated visualizations, thereby improving reasoning performance, as our ablation results confirm.
>
> We thank the reviewer for acknowledging the potential broader impact of our work to a diverse set of newly released benchmarks. We will include relevant discussion and corresponding modifications in our camera-ready version based on the valuable suggestions from the reviewer upon acceptance.

---

### Official Review · Reviewer_5f3V · 2025-03-14

**Overall Recommendation:** 4

**Summary:**

This paper presents Multimodal VoT which integrate visual generation during MLLM’s reasoning process. The idea is straight forward and the motivation is inspired from the theory about how human reasoning in both verbal and non-verbal channels. In order to increase the image generation quality, the authors proposed token discrepancy loss. The proposed framework are further fine-tuned and evaluated on three visual spatial reasoning tasks. Compared to language-based reasoning, MVoT exhibits better robustness and performance.

**Claims And Evidence:**

Yes the claims in this paper are clear and well grounded.

**Essential References Not Discussed:**

I do not identify any important but missing references in this paper. But I suggest the authors to discuss some recent and highly related work on the spatial reasoning task:

```bash
[1] Ramakrishnan, Santhosh Kumar, et al. "Does Spatial Cognition Emerge in Frontier Models?." arXiv preprint arXiv:2410.06468 (2024).
[2] Yang, J., Yang, S., Gupta, A. W., Han, R., Fei-Fei, L., & Xie, S. (2024). Thinking in space: How multimodal large language models see, remember, and recall spaces. arXiv preprint arXiv:2412.14171.
```

**Experimental Designs Or Analyses:**

The experiments look good to me.

**Methods And Evaluation Criteria:**

The proposed methods make sense and the evaluation criteria also sounds good. One only deficit is the scope of the evaluation task are synthetic and under a limited scale.

**Other Comments Or Suggestions:**

- It would be better if the authors can conduct the following experiments:
    - Evaluate MVoT on real-world visual reasoning tasks
    - Test MVoT on other abstract reasoning tasks like ARC challenge

**Other Strengths And Weaknesses:**

Strengths:

- Despite straightforward, the idea to visualize each reasoning step (especially under multimodal scope) are really interesting and worth a lot of explorations.
- Three benchmarks used in this paper is interesting, and MVoT gets competitive performance compared to other approaches.

Weaknesses:

- All three benchmarks used in this paper is synthetic and under limited scale.
- With simple fine-tuning, the models’ performance on all the three benchmarks can be boosted to over 90%, which may implies the oversimplification of these benchmarks.
- The proposed token discrepancy loss seems more like a trick to improve model’s image generation capability, but does not closely related to the reasoning process.

**Questions For Authors:**

- What’s the reason for using lora instead of fully fine-tuning?
- This paper takes a multimodal-native pre-trained model as a foundation which tokenize both the image input and output to discrete token. However, most of available and competitive multimodal models are trained with a continuous visual encoder like CLIP. Can MVoT be built on top of these models?
- RL-trained LLM show competitive reasoning capabilities. What’s the performance if benchmark reasoning LLM (like ChatGPT-o1) on three benchmarks with language only input output? (E.g., the input image can be represented as matrices or coordinates)

**Relation To Broader Scientific Literature:**

This paper is closely related to the general multimodal and reasoning fields.

**Theoretical Claims:**

This paper does not provide any theoretical proof.

---

> ### Author Rebuttal · Authors · 2025-04-01
>
> Thank you for your recognition of our work and your valuable suggestions. We would like to address your comments as follows:
>
> **Benchmark selection**
>
> > All three benchmarks used in this paper is synthetic and under limited scale.
>
> Our use of grid-based benchmarks was intentional to ensure better controllability and to systematically evaluate various aspects of spatial reasoning—from pattern complexity to action complexity. In addition, grid-based benchmarks enable easier evaluation towards whether the generated visualization is correct in terms of the spatial transition between steps rather than focusing on the image pixel details.
>
> > With simple fine-tuning, the models’ performance on all the three benchmarks can be boosted to over 90%, which may implies the oversimplification of these benchmarks.
>
> We emphasize that these benchmarks are not oversimplified. In fact, larger models consistently fail across all tasks, and even with fine-tuning, models only achieve up to 70% accuracy on FROZENLAKE. In contrast, our proposed MVoT surpasses 80% accuracy, highlighting the non-triviality of the benchmarks and the effectiveness of our approach.
>
> **Token discrepancy loss**
>
> > The proposed token discrepancy loss seems more like a trick to improve model’s image generation capability, but does not closely related to the reasoning process.
>
> We believe that there may be some misunderstanding. Our design is grounded in the hypothesis that visualization quality is closely related to reasoning performance. Experimental results across all three tasks show that more accurate visualizations consistently improve reasoning performance.  Furthermore, our ablation studies indicate that integrating the token discrepancy loss yields higher-quality visualizations with fewer redundancies. By penalising deviations from ground-truth visual tokens, it effectively guides the model toward more faithful and visually semantically aligned generated visualizations, thereby improving reasoning performance. Notably, without the token discrepancy loss, the generated visualizations struggle to enhance reasoning, underscoring that it is not a mere “trick” but an essential component for our framework. Therefore, we emphasize the relevance and contribution of the token discrepancy loss to MVoT’s overall reasoning process.
>
> **Why LoRA instead of fully fine-tuning?**
>
> We chose LoRA due to computational constraints as an approximation of full model fine-tuning. Full fine-tuning requires recording and updating gradients for all the model parameters, which is resource-intensive.
>
> **Can MVoT be built on top of other multimodal models with continuous visual encoder?**
>
> We appreciate the insightful question regarding broader applicability. In principle, MVoT can be extended to other multimodal models that support interleaved text-image generation. However, many current MLLMs with continuous visual encoders are restricted to producing textual outputs only. We look forward to the development of architectures that support richer, interleaved multimodal outputs, which would further broaden the applicability of MVoT.
>
> **Language-only input output?**
>
> We agree that it would be interesting to see how RL helps in textual spatial reasoning. However, our current work specifically targets multimodal spatial reasoning, and extending to language-only reasoning lies out of the present scope. We consider it a promising avenue for future exploration.
>
> **Evaluation of MVoT on real-world visual reasoning tasks and other abstract reasoning tasks.**
>
> We agree that it would be interesting to see how MVoT performs on these tasks. However, due to the current lack of interleaved text-image reasoning datasets in those domains, we focused on tuning Anole with LoRA on three grid-based datasets. This constraint limits generalizability, but we hope our work offers foundational insights and inspires further studies in this direction.
>
> Once again, we thank the reviewers for their constructive comments and for pointing us toward relevant recent work. We will reflect these considerations and include the suggested references in our camera-ready submission upon acceptance.

---

### Decision · Program_Chairs · 2025-05-01

**Decision:**

Accept (poster)

**Comment:**

This paper introduces Multimodal Visualization-of-Thought (MVoT), a novel multimodal reasoning paradigm that extends the chain-of-thought (CoT) concept by interleaving textual reasoning with visual outputs. Specifically, by introducing a token discrepancy loss, the authors demonstrate that MVoT can enhance complex spatial reasoning tasks. Experiments on three synthetic grid-based benchmarks (e.g., MAZE, MiniBehavior, FrozenLake) indicate that this new multimodal reasoning paradigm can achieve competitive performance.

Overall, all reviewers recognize the novelty of this paradigm and the solid experimental support provided. But meanwhile, they raise multiple major concerns, including: 1) all current evaluations are in-distribution, therefore it is unclear whether MVoT can generalize to more realistic or out-of-distribution scenarios; 2) its scalability to more complex scenes is a concern; 3) the intuition behind introducing the token discrepancy loss is not sufficiently clear, and its technical novelty is viewed as limited; and 4) it is unclear whether MVoT can generalize to other multimodal LLMs with a continuous visual encoder like CLIP.

The authors' rebuttal is responsive and effectively addresses most of these concerns. As a result, all reviewers unanimously vote to accept the submission, and the AC concurs with this decision.